# Predictability Enables Parallelization
# of Nonlinear State Space Models

**Xavier Gonzalez**[*]
Stanford University
xavier18@stanford.edu

**Leo Kozachkov**[*†]
IBM Research
leokoz8@brown.edu

**David M. Zoltowski**
Stanford University
dzoltow@stanford.edu

**Kenneth L. Clarkson**
IBM Research
klclarks@us.ibm.com

**Scott W. Linderman**
Stanford University
scott.linderman@stanford.edu

## Abstract

The rise of parallel computing hardware has made it increasingly important to understand which nonlinear state space models can be efficiently parallelized. Recent advances like DEER [1] and DeepPCR [2] recast sequential evaluation as a parallelizable optimization problem, sometimes yielding dramatic speedups. However, the factors governing the difficulty of these optimization problems remained unclear, limiting broader adoption. In this work, we establish a precise relationship between a system's dynamics and the conditioning of its corresponding optimization problem, as measured by its Polyak-Łojasiewicz (PL) constant. We show that the predictability of a system, defined as the degree to which small perturbations in state influence future behavior and quantified by the largest Lyapunov exponent (LLE), impacts the number of optimization steps required for evaluation. For predictable systems, the state trajectory can be computed in at worst $\mathcal{O}((\log T)^2)$ time, where $T$ is the sequence length: a major improvement over the conventional sequential approach. In contrast, chaotic or unpredictable systems exhibit poor conditioning, with the consequence that parallel evaluation converges too slowly to be useful. Importantly, our theoretical analysis shows that predictable systems always yield well-conditioned optimization problems, whereas unpredictable systems lead to severe conditioning degradation. We validate our claims through extensive experiments, providing practical guidance on when nonlinear dynamical systems can be efficiently parallelized. We highlight predictability as a key design principle for parallelizable models.

## 1 Introduction

Parallelization has been central to breakthroughs in deep learning, with GPUs enabling the fast training of large neural networks. In contrast, nonlinear state space models like recurrent neural networks (RNNs) have resisted efficient parallelization on GPUs due to their sequential nature.

Recent work addresses this mismatch by reformulating sequential dynamics into parallelizable optimization problems. Notably, the DEER/DeepPCR algorithm [1, 2] evaluates nonlinear state space dynamics by minimizing a residual-based merit function, facilitating efficient parallel computation

---

[*]Equal contribution.
[†]Now at Brown University.

39th Conference on Neural Information Processing Systems (NeurIPS 2025).

via the Gauss-Newton method.[3] Gonzalez et al. [3] further developed these methods, including quasi-Newton methods and trust-region methods for parallel evaluation of nonlinear dynamical systems. These methods evaluate nonlinear dynamical systems by iteratively linearizing the nonlinear system and evaluating the resulting linear dynamical system (LDS) with a parallel (a.k.a. associative) scan [4, 5]. Each parallel evaluation of an LDS implements one optimization step [1–3, 6].

The usefulness of this optimization-based reformulation depends on two key factors: (a) the computational time per optimization step, and (b) the number of optimization steps required. The computational time per optimization step is only logarithmic in the sequence length, thanks to its parallel structure. However, the number of steps is governed by the conditioning of the merit function, and that remains poorly understood. In this paper, we characterize the merit function's conditioning, allowing us to draw a sharp distinction between systems that are amenable to efficient parallelization via merit function minimization and those that are not (see Figure 1, which is generated from trajectories of an RNN). Geometrically, we show that unpredictable systems lead to merit functions that have regions of extreme flatness, which can lead to very slow convergence.

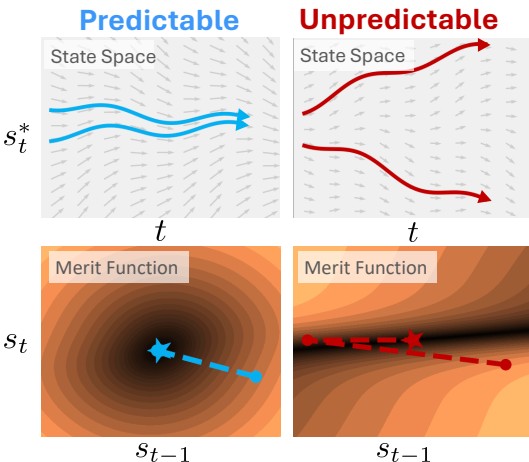

**Figure 1:** Predictable nonlinear state space models can be recast as well-conditioned, parallelizable optimization problems.

Drawing from nonlinear dynamical systems theory—particularly contraction analysis [7] and Lyapunov exponent methods [8]—we formalize the relationship between system predictability and the conditioning of the merit function. **Unpredictable systems** are dynamical systems whose future behavior is highly sensitive to small perturbations. A common example is a chaotic system, like the weather: a butterfly flapping its wings in Tokyo today can lead to a thunderstorm in Manhattan next month [9, 10]. By contrast, **predictable systems** [11, 12] are those in which small perturbations are "forgotten." A familiar example is aviation: a patch of choppy air rarely makes an airplane land at the wrong airport. A more formal definition of (un)predictability is given in Definition 1. Our results establish key theoretical principles, make connections between optimization theory and dynamical systems, and demonstrate the practical applicability of parallel computations across a wide range of nonlinear state space modeling tasks.

**Contributions & Outline**   Our central finding is that predictable systems give rise to well-conditioned merit functions, making them amenable to efficient parallelization. Unpredictable (e.g., chaotic) systems produce poorly conditioned merit functions and are not easily parallelizable.

The paper is organized as follows. Section 2 provides background, with formal definitions of predictable and unpredictable nonlinear state space models. Section 3 presents two key theoretical results that characterize the conditioning of the merit function, showing that the Polyak–Łojasiewicz (PL) constant $\mu$ of the merit function is controlled by the predictability of the dynamics (Theorem 2), and that the Lipschitz constant of the residual function Jacobian is governed by the nonlinearity of the dynamics (Theorem 3). Section 4 then uses the results about the conditioning of the merit function to prove results about Gauss-Newton in particular. We prove global linear rates of convergence for Gauss-Newton, with the precise rate scaling with the unpredictability of the problem (Theorem 4), and we characterize the basin of quadratic convergence in terms of the predictability and nonlinearity of the underlying dynamics (Theorem 5). In Section 5 we illustrate our results with experiments, and in Section 6 we conclude by summarizing context, implications, limitations, and future directions.

---

[3]DEER [1] and DeepPCR [2] were concurrent works that both proposed to use the Gauss-Newton method for optimizing nonlinear sum of squares to parallelize sequential processes. In this paper, we therefore use DEER, DeepPCR, and Gauss-Newton interchangeably.

## 2 Problem Statement & Background

**Notation** Throughout the paper, we use $T$ to denote the length of a sequence and $D$ to represent the dimensionality of a nonlinear state space model. Elements in $\mathbb{R}$, $\mathbb{R}^D$ or $\mathbb{R}^{D \times D}$ are written using non-bold symbols, while elements in $\mathbb{R}^{TD}$ or $\mathbb{R}^{TD \times TD}$ are denoted with bold symbols.

**Sequential Evaluation vs. Merit Function Optimization** We consider the $D$-dimensional nonlinear state space model

$$s_t = f_t(s_{t-1}) \in \mathbb{R}^D. \tag{1}$$

A simple example is an input-driven nonlinear RNN, $s_t = \tanh(W s_{t-1} + B u_t)$, where $W$ and $B$ are weight matrices and $u_t$ is the input into the network at time $t$. We want to compute the state trajectory, $(s_1, \ldots, s_T)$, starting from an initial condition $s_0$, for a given sequence of functions $f_1, \cdots, f_T$.

Systems of the form (1) are widespread across science and engineering. Examples include physics (numerical weather prediction, molecular dynamics), biology (gene regulatory networks, population dynamics), engineering (control, robotics), and economics (macroeconomic forecasting, asset pricing). In machine learning, sequential operations arise in recurrent neural networks, iterative optimization, and the sampling pass of a diffusion model [2, 13]. Sequential operations even appear in the problem of evaluating transformer blocks over depth [14–19]. In probabilistic modeling, sequential operations arise in Markov Chain Monte Carlo [20]. In all of these cases, the state evolves through nonlinear transformations that capture the system's underlying dynamics.

The obvious approach is to sequentially compute the states according to eq. (1), taking $T$ steps. Alternatively, one can cast state evaluation as an optimization problem. While less intuitive, an advantage of this approach is that it admits parallel computation [1–3]. Depending on the properties of the nonlinear state space model, the optimization algorithm, and the available hardware, the latter approach can be significantly faster than sequential evaluation.

We define the residual and corresponding merit[4] function $\mathcal{L}$ by stacking the elements $s_t \in \mathbb{R}^D$ of a trajectory into a $TD$-dimensional vector $\mathbf{s}$ and considering the vector of temporal differences,

$$\mathbf{r}(\mathbf{s}) := \text{vec}\left([s_1 - f_1(s_0), \ldots, s_T - f_T(s_{T-1})]\right) \in \mathbb{R}^{TD}, \qquad \mathcal{L}(\mathbf{s}) := \frac{1}{2} \|\mathbf{r}(\mathbf{s})\|_2^2, \tag{2}$$

where $\text{vec}(\cdot)$ denotes the flattening of a sequence of vectors into a single column vector. The true trajectory $\mathbf{s}^*$ is then obtained by minimizing $\mathcal{L}(\mathbf{s})$. Note that the residual is zero only at the true trajectory, i.e., when $s_1, s_2, \cdots, s_T$ satisfy (1) at every time point, so $\mathbf{s}^*$ is the unique global minimum of $\mathcal{L}(\mathbf{s})$.

DeepPCR [2] and DEER [1] minimize the merit function using Gauss–Newton updates. Each update takes the form

$$\mathbf{s}^{(i+1)} = \mathbf{s}^{(i)} - \mathbf{J}(\mathbf{s}^{(i)})^{-1} \mathbf{r}(\mathbf{s}^{(i)}). \tag{3}$$

where $\mathbf{J}(\mathbf{s}^{(i)})$ denotes the Jacobian of the residual function, evaluated at the current iterate $\mathbf{s}^{(i)}$. The Jacobian is a $TD \times TD$ matrix with $D \times D$ block bidiagonal structure

$$\mathbf{J}(\mathbf{s}^{(i)}) := \frac{\partial \mathbf{r}}{\partial \mathbf{s}}(\mathbf{s}^{(i)}) = \begin{pmatrix} I_D & 0 & \ldots & 0 & 0 \\ -J_2^{(i)} & I_D & \ldots & 0 & 0 \\ \vdots & \vdots & \ddots & \vdots & \vdots \\ 0 & 0 & \ldots & I_D & 0 \\ 0 & 0 & \ldots & -J_T^{(i)} & I_D \end{pmatrix} \quad \text{where} \quad J_t^{(i)} := \frac{\partial f_t}{\partial s_{t-1}}(s_{t-1}^{(i)}). \tag{4}$$

Due to this block bidiagonal structure, solving $\mathbf{J}(\mathbf{s}^{(i)})^{-1} \mathbf{r}(\mathbf{s}^{(i)})$ amounts to solving a linear recursion, which can be done in $\mathcal{O}(\log T)$ time with a parallel scan [1, 3, 5, 22, 23]. Further details are given in Appendix A.

This sublinear time complexity per step is only useful if the number of optimization steps required to minimize the merit function is small, otherwise it would be more efficient to evaluate the recursion sequentially. Thus, we seek to characterize the conditioning of the merit function — determining

---

[4]While minimizing a "merit function" is admittedly counterintuitive, we follow Nocedal and Wright [21, see eq. 11.35] in this convention.

when it is well-conditioned and when it is not — since this affects the difficulty of finding its minimum. Equation (4) already offers an important clue. The presence of the nonlinear state-space model Jacobians $J_t$, which measure the local stability and predictability of the nonlinear dynamics, foreshadows our central finding: the system's predictability dictates the conditioning of the merit function.

**Predictable Systems: Lyapunov Exponents and Contraction** Predictability is usually defined through its antonym: *un*predictability [9, 10]. In an unpredictable system, the system's intrinsic sensitivity amplifies small perturbations and leads to massive divergence of trajectories. Predictable systems show the opposite behavior: small perturbations are diminished over time, rather than amplified. The notion of (un)predictability can be formalized through various routes such as chaos theory [24, 25] and contraction analysis [7, 26].

The definition of predictability comes from the Largest Lyapunov Exponent (LLE) [8, 10]:

---

**Definition 1** (**Predictability and Unpredictability**). Consider a sequence of Jacobians, $J_1, J_2, \cdots J_T$. We define the associated Largest Lyapunov Exponent (LLE) to be

$$\text{LLE} := \lim_{T \to \infty} \frac{1}{T} \log \left( \| J_T J_{T-1} \cdots J_1 \| \right) = \lambda, \tag{5}$$

where $\| \cdot \|$ is an induced operator norm. If $\lambda < 0$, we say that the nonlinear state space model is *predictable* at $s_0$. Otherwise, we say it is *unpredictable*.

---

Suppose we wish to evaluate the nonlinear state space model (1) from an initial condition $s_0$, but we only have access to an approximate measurement $s_0'$ that differs slightly from the true initial state. If the system is unpredictable ($\lambda > 0$), then the distance between nearby trajectories grows as

$$\| s_t - s_t' \| \sim e^{\lambda t} \| s_0 - s_0' \|. \tag{6}$$

Letting $\Delta$ denote the maximum acceptable deviation beyond which we consider the prediction to have failed, the time horizon over which the prediction remains reliable scales as

$$\text{Time to degrade to } \Delta \text{ prediction error} \sim \frac{1}{\lambda} \log \left( \frac{\Delta}{\| s_0 - s_0' \|} \right). \tag{7}$$

This relationship highlights a key limitation in unpredictable systems: even significant improvements in the accuracy of the initial state estimate yield only logarithmic gains in prediction time. The system's inherent sensitivity to initial conditions overwhelms any such improvements. Predictable systems, such as contracting systems, have the opposite property: trajectories initially separated by some distance will eventually converge towards one another (Figure 1), *improving* prediction accuracy over time.

## 3 Conditioning of Merit Function Depends on Predictability of Model

The number of optimization steps required to minimize the merit function (2) is impacted by its conditioning, which in our setting is determined by the smallest singular value of the residual function Jacobian. As we will see, what determines the smallest singular value of the residual function Jacobian is the stability, or predictability, of the underlying nonlinear state space model (1).

### 3.1 The Merit Function is PL

To begin, we show that the merit function (2) satisfies the Polyak-Łojasiewicz (PL) condition [27, 28], also known as the gradient dominance condition [29]. A function $\mathcal{L}(\mathbf{s})$ is $\mu$-PL if it satisfies, for $\mu > 0$,

$$\frac{1}{2} \| \nabla \mathcal{L}(\mathbf{s}) \|^2 \geq \mu \left( \mathcal{L}(\mathbf{s}) - \mathcal{L}(\mathbf{s}^*) \right) \tag{8}$$

for all $\mathbf{s}$. The largest $\mu$ for which eq. (8) holds for all $\mathbf{s}$ is called the PL constant of $\mathcal{L}(\mathbf{s})$.

**Proposition 1.** *The merit function $\mathcal{L}(\mathbf{s})$ defined in eq. (2) satisfies eq. (8) for*

$$\mu := \inf_{\mathbf{s}} \sigma_{\min}^2(\mathbf{J}(\mathbf{s})). \tag{9}$$

*Proof.* See Appendix B. This result, known in the literature for general sum-of-squares [30], is included here for context and completeness. $\square$

Proposition 1 is important as it characterizes the *flatness* of the merit function. If $\mu$ is very small in a certain region, this indicates that the norm of the gradient can be very small in that region, which can make gradient-based optimization inefficient. Proposition 1 also links $\sigma_{\min}(\mathbf{J})$—important for characterizing the conditioning of $\mathbf{J}$—to the geometry of the merit function landscape.

### 3.2 Merit Function PL Constant is Controlled by the Largest Lyapunov Exponent of Dynamics

As stated earlier, the Largest Lyapunov Exponent is a commonly used way to define the (un)predictability of a nonlinear state space model. In order to proceed, we need to control more carefully how the product of Jacobian matrices in (5) behaves for finite-time products. We will assume that there exists a "burn-in" period where the norm of Jacobian products can transiently differ from the LLE. In particular, we assume that

$$\forall t > 1, \ \forall k \geq 0, \ \forall \mathbf{s}, \qquad b\, e^{\lambda k} \leq \|J_{t+k-1} J_{t+k-2} \cdots J_t\| \leq a\, e^{\lambda k}, \tag{10}$$

where $a \geq 1$ and $b \leq 1$. The constant $a$ quantifies the potential for transient growth—or overshoot—in the norm of Jacobian products before their long-term behavior emerges, while $b$ quantifies the potential for undershoot.

**Theorem 2.** *Assume that the LLE regularity condition* (10) *holds. Then the PL constant $\mu$ satisfies*

$$\frac{1}{a} \cdot \frac{e^{\lambda} - 1}{e^{\lambda T} - 1} \leq \sqrt{\mu} \leq \min\left( \frac{1}{b} \cdot \frac{1}{e^{\lambda(T-1)}}, 1 \right). \tag{11}$$

*Proof.* See Appendix C for the full proof and discussion. We provide a brief sketch. Because $\sigma_{\min}(\mathbf{J}) = 1/\sigma_{\max}(\mathbf{J}^{-1})$, it suffices to control $\|\mathbf{J}^{-1}\|_2$. We can write $\mathbf{J} = \mathbf{I} - \mathbf{N}$ where $\mathbf{N}$ is a nilpotent matrix. Thus, it follows that $\mathbf{J}^{-1} = \sum_{k=0}^{T-1} \mathbf{N}^k$. As we discuss further in Appendix C, the matrix powers $\mathbf{N}^k$ are intimately related to the dynamics of the system. The upper bound on $\|\mathbf{J}^{-1}\|_2$ follows after applying the triangle inequality and the formula for a geometric sum. The lower bound follows from considering $\|\mathbf{N}^{T-1}\|_2$. $\square$

Theorem 2 is our main result, offering a novel connection between the predictability $\lambda$ of a nonlinear state space model and the conditioning $\mu$ of the corresponding merit function, which affects whether the system can be effectively parallelized. If the underlying dynamics are unpredictable ($\lambda > 0$), then the merit function quickly becomes poorly conditioned with increasing $T$, because the denominators of both the lower and upper bounds explode due to the exponentially growing factor. Predictable dynamics $\lambda < 0$ lead to good conditioning of the optimization problem, and parallel methods based on merit function minimization can be expected to perform well in these cases.

The proof mechanism we have sketched upper and lower bounds $\|\mathbf{J}^{-1}\|_2$ in terms of norms of Jacobian products. We only use the assumption in eq. (10) to express those bounds in terms of $\lambda$. As we discuss at length in Appendix C, we can use different assumptions from eq. (10) to get similar results. Theorem 2 and its proof should be thought of as a framework, where different assumptions (which may be more or less relevant in different settings) can be plugged in to yield specific results.

**Why Unpredictable Systems have Excessively Flat Merit Functions** Theorem 2 demonstrates that the merit function becomes extremely flat for unpredictable systems and long trajectories. This flatness poses a fundamental challenge for *any* method that seeks to compute state trajectories by minimizing the merit function. We now provide further intuition to explain why unpredictability in the system naturally leads to a flat merit landscape.

Suppose that we use an optimizer to minimize the merit function (2) for an unpredictable system until it halts with some precision. Let us further assume that the first state of the output of this optimizer following the initial condition is $\epsilon$-close to the true first state, $\|s_1 - s_1^*\| = \epsilon$. Suppose also that the residuals for all times greater than one are precisely zero—in other words, the optimizer starts with a "true" trajectory starting from initial condition $s_1$. Then the overall residual norm is at most $\epsilon$,

$$\|\mathbf{r}(\mathbf{s})\|^2 = \|s_1 - f(s_0)\|^2 \leq (\|s_1 - s_1^*\| + \|s_1^* - f(s_0)\|)^2 = \|s_1 - s_1^*\|^2 = \epsilon^2.$$

However, since $s_t$ and $s_t^*$ are by construction both trajectories of an unpredictable system starting from slightly different initial conditions $s_1$ and $s_1^*$, the distance between them will grow exponentially as a consequence of eq. (7). By contrast, predictable systems will have errors that shrink exponentially. This shows that changing the initial state $s_1$ by a small amount can lead to a massive change in the trajectory of an unpredictable system, but a *tiny change* in the merit function. Geometrically, this corresponds to the merit function landscape for unpredictable systems having excessive flatness around the true solution (Figure 1, bottom right panel). Predictable systems do not exhibit such flatness, since small residuals imply small errors. Theorem 2 formalizes this idea.

### 3.3 Residual function Jacobian Inherits the Lipschitzness of the Nonlinear State Space Model

In addition to the parameter $\mu$, which measures the conditioning of the merit function, the difficulty of minimizing the merit function is also influenced by the Lipschitz continuity of its Jacobian $\mathbf{J}$. The following theorem establishes how the Lipschitz continuity of the underlying sequence model induces Lipschitz continuity in $\mathbf{J}$.

**Theorem 3.** *If the dynamics of the underlying nonlinear state space model have L-Lipschitz Jacobians, i.e.,*

$$\forall t > 1, \quad s, s' \in \mathbb{R}^D : \quad \|J_t(s) - J_t(s')\| \leq L\|s - s'\|,$$

*then the residual function Jacobian $\mathbf{J}$ is also L-Lipschitz, with the same L.*

*Proof.* See Appendix D. $\qquad\qquad\square$

Theorem 3 will be important for the analysis in Section 4, where we consider convergence rates. Because Gauss-Newton methods rely on iteratively linearizing the dynamics (or equivalently the residual), they converge in a single step for linear dynamics $L = 0$, and converge more quickly if the system is close to linear ($L$ is closer to 0).

## 4  Rates of Convergence for Optimizing the Merit Function

In Section 3, we established that the predictability of the nonlinear state space model directly influences the conditioning of the merit function. This insight is critical for analyzing *any* optimization method used to compute trajectories via minimization of the merit function.

In this section, we apply those results to study the convergence behavior of the Gauss-Newton (DEER) algorithm for the merit function defined in eq. (2). See Appendix A for a brief overview of DEER. We derive worst-case bounds on the number of optimization steps required for convergence. In addition, we present an average-case analysis of DEER that is less conservative than the worst-case bounds and more consistent with empirical observations.

**DEER Always Converges Globally at a Linear Rate**  Although DEER is based on the Gauss-Newton method, which generally lacks global convergence guarantees, we prove that DEER always converges globally at a linear rate. This result relies on the problem's specific hierarchical structure, which ensures that both the residual function Jacobian $\mathbf{J}$ and its inverse are lower block-triangular. In particular we prove the following theorem

**Theorem 4.** *Let the DEER (Gauss–Newton) updates be given by eq. (3), and let $\mathbf{s}^{(i)}$ denote the i-th iterate. Let $\mathbf{e}^{(i)} := \mathbf{s}^{(i)} - \mathbf{s}^*$ denote the error at iteration i, and assume the regularity condition in eq. (10). Then the error converges to zero at a linear rate:*

$$\|\mathbf{e}^{(i)}\|_2 \leq \chi_w\, \beta^i \|\mathbf{e}^{(0)}\|_2,$$

*for some constant $\chi_w \geq 1$ independent of i, and a convergence rate $0 < \beta < 1$.*

*Proof.* See Appendix E. $\qquad\qquad\square$

Theorem 4 is unexpected since, in general, Gauss-Newton methods do not enjoy global convergence. The key caveat of this theorem is the multiplicative factor $\chi_w$, which can grow exponentially with the sequence length $T$. This factor governs the extent of transient error growth before the decay term $\beta^i$ eventually dominates.

Theorem 4 has several useful, practical consequences. First, when the nonlinear state space model is sufficiently contracting ($\lambda$ is sufficiently negative), then $\chi_w$ in Theorem 4 can be made small, implying that in this case DEER converges with little-to-no overshoot (Appendix F).

Theorem 4 also lets us establish key worst-case and average-case bounds on the number of steps needed for Gauss-Newton to converge to within a given distance of the solution. In particular, when $\chi_w$ does not depend on the sequence length $T$, then Theorem 4 implies Gauss-Newton will only require $\mathcal{O}\left((\log T)^2\right)$ total computational time, with one $\log$ factor coming from the parallel scan at each optimization step and the other coming from the total number of optimization steps needed. We elaborate on these points in Appendix G.

**Size of DEER Basin of Quadratic Convergence**    It is natural that DEER depends on the Lipschitzness of $\mathbf{J}$ since Gauss-Newton converges *in one step* for linear problems, where $L = 0$. In Section 3, we showed that the conditioning of the merit function, as measured by the PL-constant $\mu$, depends on the stability, or predictability, of the nonlinear dynamics. Thus, the performance of DEER depends on the ratio of the nonlinearity and stability of the underlying nonlinear state space model. Note that once $\mathbf{s}$ is inside the basin of quadratic convergence, it takes $O(\log\log(1/\epsilon))$ steps to reach $\epsilon$ residual (effectively a constant number of steps).

**Theorem 5.** *Let $\mu$ denote the PL-constant of the merit function, which Theorem 2 relates to the LLE $\lambda$. Let $L$ denote the Lipschitz constant of the Jacobian of the dynamics function $J(s)$. Then, $2\mu/L$ lower bounds the radius of the basin of quadratic convergence of DEER; that is, if*

$$||\mathbf{r}(\mathbf{s}^{(i)})||_2 < \frac{2\mu}{L}, \tag{12}$$

*then $\mathbf{s}^{(i)}$ is inside the basin of quadratic convergence. In terms of the LLE $\lambda$, it follows that if*

$$||\mathbf{r}(\mathbf{s}^{(i)})||_2 < \frac{2}{a^2 L} \cdot \left(\frac{e^\lambda - 1}{e^{\lambda T} - 1}\right)^2,$$

*then $\mathbf{s}^{(i)}$ is inside the basin of quadratic convergence.*

*Proof.* See Appendix H. We make no claim about the originality of lower bounding the size of the basin of quadratic convergence in Gauss-Newton. In fact, our proof of Theorem 5 closely follows the convergence analysis of *Newton's* method in Section 9.5.3 of Boyd and Vandenberghe [31]. Our contribution is we highlight the elegant way the predictability $\lambda$ and nonlinearity $L$ of a dynamical system influence an important feature of its merit function's landscape. $\square$

## 5    Experiments

We conduct experiments to support the theory developed above, demonstrating that predictability enables parallelization of nonlinear SSMs. To illustrate this point, we use Gauss-Newton optimization (aka DEER [1, 2]). We provide more experimental details in Appendix K. Our code is at https://github.com/lindermanlab/predictability_enables_parallelization

**The Convergence Rate Exhibits a Threshold between Predictable and Chaotic Dynamics**
Theorem 2 predicts a sharp phase transition in the conditioning of the merit function at $\lambda = 0$, which should be reflected in the number of optimization steps required for convergence. To empirically validate this prediction, we vary both the LLE and sequence length $T$ within a parametric family of recurrent neural networks (RNNs), and measure the number of steps DEER takes to converge. We generate mean-field RNNs following Engelken et al. [32], scaling standard normal weight matrices by a single parameter that controls their variance and therefore the expected LLE. In Figure 2, we observe a striking correspondence between the conditioning of the optimization problem (represented by $-\log \tilde{\mu}$, where $\tilde{\mu}$ is the lower bound for $\mu$ from Theorem 2) and the number of steps DEER takes to converge. This relationship holds across the range of LLEs, $\lambda$, and sequence lengths, $T$. There is a rapid threshold phenomenon around $\lambda = 0$, which divides predictable from unpredictable dynamics, precisely as expected from Theorem 2. As we discuss in Appendix K.1, the correspondence between $-\log \tilde{\mu}$ and the number of optimization steps needed for convergence can be explained by DEER iterates approaching the basin of quadratic convergence with linear rate.

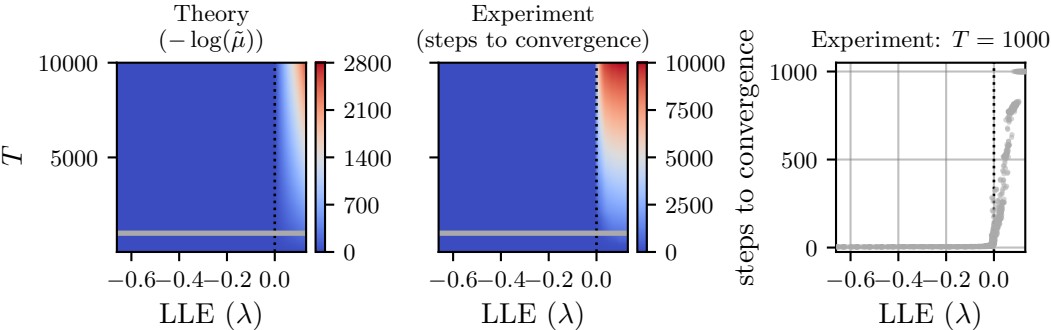

**Figure 2: Threshold phenomenon in DEER convergence based on system predictability.** In a family of RNNs, DEER has fast convergence for predictable systems and prohibitively slow convergence for chaotic systems. **Left (Theory):** We depict Theorem 2, illustrating how the conditioning of the optimization problem degrades as $T$ and the LLE ($\lambda$) increase. **Center (Experiment):** We vary $\lambda$ across the family of RNNs, and observe a striking concordance in the number of DEER optimization steps empirically needed for convergence with our theoretical characterization of the conditioning of the optimization problem. **Right:** For 20 seeds, each with 50 different values of $\lambda$, we plot the relationship between $\lambda$ and the number of DEER steps needed for convergence for the sequence length $T = 1000$ (gray line in left and center panels). We observe a sharp increase in the number of optimization steps at precisely the transition between predictability and unpredictability.

In Appendix K.3, we provide additional experiments in this setting. We parallelize the sequential rollout with other optimizers like quasi-Newton and gradient descent, and observe that the number of steps these optimizers take to converge also scales with the LLE. We also record wallclock times on an H100, and observe that DEER is faster than sequential by an order of magnitude in predictable settings, but slower by an order of magnitude in unpredictable settings.

**DEER can converge quickly for predictable trajectories passing through unpredictable regions**
DEER may still converge quickly even if the system is unpredictable in certain regions. As long as the system is predictable on average, as indicated by a negative LLE, DEER can still converge quickly. This phenomenon is why we framed Theorem 2 in terms of the LLE $\lambda$ and burn-in constants $a$, as opposed to a weaker result that assumes the system Jacobians have singular values less than one over the entire state space (see our discussion of condition (10) vs. condition (22) in Appendix C).

To illustrate, we apply DEER to Langevin dynamics in a two-well potential (visualized in Figure 3 for $D = 2$). The dynamics are stable within each well but unstable in the region between them. Despite this local instability, the system's overall behavior is governed by time spent in the wells, resulting in a negative LLE and sublinear growth in DEER's convergence steps with sequence length $T$ (Figure 3, right subplot). Additional details and discussion are in Appendix K.4.

Notably, prior works such as Lim et al. [1] and Gonzalez et al. [3] initialized optimization from $\mathbf{s}^{(0)} = \mathbf{0}$, which lies entirely in the unstable region. Thus, our theoretical insights into predictability and parallelizability suggest practical improvements for initialization.

**Application: Chaotic Observers** Finally, we demonstrate a practical application of our theory in the efficient parallelization of chaotic observers. Observers are commonly used to reconstruct the full state of a system from partial measurements [33, 34]. On nine chaotic flows from the `dysts` benchmark dataset [35], Table 1 shows that while DEER converges prohibitively slowly on chaotic systems, it converges rapidly on stable observers of these systems, in accordance with our theory that predictability implies parallelizability. For more details, see Appendix K.5.

## 6  Discussion

Recent work demonstrated that parallel computing hardware like GPUs can be used to rapidly compute state trajectories of nonlinear state space models (nSSMs) by recasting the trajectory as the solution to an optimization problem. In this work, we provide the first precise characterization of

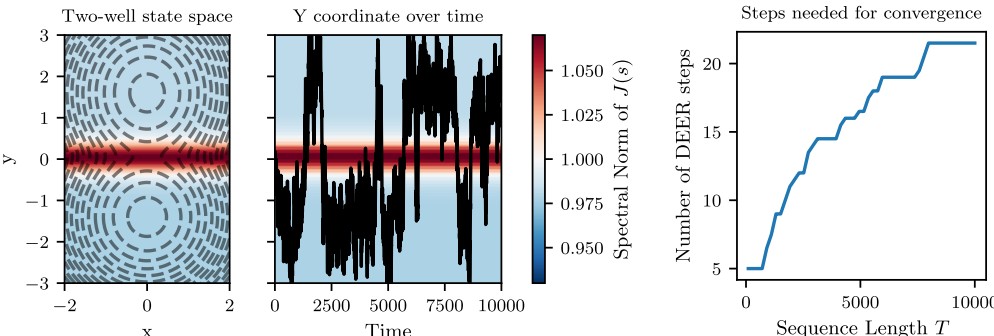

**Figure 3:** DEER converges quickly for Langevin dynamics in a two-well potential. **(Left)** An illustration of the two-well potential state space in $D = 2$. We superimpose a contour plot of the potential on a color scheme showing the spectral norm of the dynamics Jacobian (blue indicates stability, red instability). **(Center)** A trace plot for the $y$-coordinate. The LLE of the system is $-0.0145$. **(Right)** We observe that this system, which has negative LLE, enjoys sublinear scaling in the sequence length $T$ in the number of DEER iterations needed to converge. We plot the median number of DEER steps to convergence over 20 random seeds.

**Table 1:** Comparison of system and observer LLEs and number of DEER steps for $T = 30,000$ and Euler discretization step size $\Delta t = 0.01$.

| System | LLE (System) | LLE (Observer) | DEER Steps (System) | DEER Steps (Observer) |
|---|---|---|---|---|
| ABC | 0.16 | -0.08 | 4243 | 3 |
| Chua's Circuit | 0.02 | -1.37 | 697 | 14 |
| Kawczynski-Strizhak | 0.01 | -3.08 | 29396 | 2 |
| Lorenz | 1.02 | -6.28 | 30000 | 3 |
| Nosé–Hoover Thermostat | 0.02 | -0.13 | 29765 | 3 |
| Rössler | 0.01 | -0.07 | 29288 | 7 |
| SprottB | 0.20 | -0.39 | 29486 | 2 |
| Thomas | 0.01 | -3.07 | 12747 | 7 |
| Vallis El Niño | 0.58 | -2.48 | 30000 | 3 |

the optimization problem's inherent difficulty, which determines if parallelization will be faster in practice than sequential evaluation. We show that the conditioning of the optimization problem is governed by the predictability of the underlying dynamics. We translate this insight into worst-case performance guarantees for specific optimizers, including Gauss–Newton (DEER). Our main takeaway is: *Predictable dynamics yield well-conditioned merit functions, enabling rapid convergence. Unpredictable dynamics produce flat or ill-conditioned merit landscapes, resulting in slow convergence or numerical failure.*

**Related Work** While Lim et al. [1] and Danieli et al. [2] introduced parallel Newton methods, they did not prove their global convergence. Gonzalez et al. [3] proved global convergence, though only with worst-case bounds of $T$ optimization steps. These prior works did not address the relationship between system dynamics and conditioning, or establish global linear convergence rates.

Global convergence rates for Gauss-Newton are rare, despite the breadth of optimization literature [21, 31, 36, 37]. Theorem 4 establishes global convergence with linear rate for Gauss-Newton by leveraging our specific problem structure, though similar results have existed for *local* linear convergence [38], most famously the Newton-Kantorovich theorem [39].

Fifty years ago, Hyafil and Kung [40] and Kung [41] showed that linear recursions enjoy speedups from parallel processors while nonlinear recursions of rational functions with degree larger than one cannot. These prescient works set the stage for our more general findings, which explicitly link the dynamical properties of the recursion to its parallelizability. Parallel-in-time methods for continuous

systems also have a long history [42–44], with Chartier and Philippe [45] showing that dissipative systems can be parallelized using multiple shooting. Furthermore, Danieli and MacLachlan [46] and De Sterck et al. [47] study the CFL number for determining the usefulness of multigrid systems. Connecting this work with our paper is an interesting direction for future research.

More recently, several works have parallelized diffusion models via fixed-point iteration, including worst-case guarantees of $T$ steps [13, 48, 49] as well as polylogarithmic rates in $T$ [50, 51]. Lu et al. [52] develops quasi-Newton methods for sampling from diffusion models and, like us, shows a two-phase model of linear followed by quadratic convergence. Crucially, prior work has not focused on the merit function, which we can define for any discrete-time dynamical system and optimizer.

To our knowledge, no prior work connects the LLE of a dynamical system to the conditioning of the corresponding optimization landscape, as established in Theorem 2. In particular, we showed that systems with high unpredictability yield poorly conditioned (i.e., flat) merit functions, linking dynamical instability to optimization difficulty in a geometrically appealing way.

The centrality of parallel sequence modeling architectures like transformers [53], deep SSMs [54, 23, 55], and linear RNNs [56] in modern machine learning underscores the need for our theoretical work. Merrill et al. [57] explored the question of parallelizability through the lens of circuit complexity, analyzing when deep learning models can solve structured tasks in constant depth. Their focus complements ours, and suggests an opportunity for synthesis in future work [58].

**Implications**   Our work unlocks three key implications for nonlinear state space models.

First, it provides a principled way to determine, *a priori*, whether optimization-based parallelization of a given model is practical. In many robotic or control systems, particularly ones that are strongly dissipative, this insight can enable orders-of-magnitude speed-ups on GPUs [59–67].

For example, the concurrent work of Zoltowski et al. [20] developed and leveraged quasi-Newton methods to parallelize Markov Chain Monte Carlo over the sequence length, attaining order-of-magnitude speed-ups. These speed-ups occurred because the quasi-Newton methods converged quickly in the settings considered. Suggestively, MCMC chains are contractive in many settings [68–70]. A precise characterization of what makes an MCMC algorithm and target distribution predictable would provide useful guidance for when one should aim to parallelize MCMC over the sequence length. Providing precise theoretical justification for parallelizing MCMC over the sequence length is an exciting avenue for future work.

Second, our results impact architecture *design*. When constructing nonlinear dynamical systems in machine learning—such as novel RNNs—parallelization benefits are maximized when the system is made predictable. Given the large body of work on training stable RNNs [71–82], many effective techniques already exist for enforcing stability or predictability during training. A common approach is to *parameterize* the model's weights so that the model is always stable (see Appendix I).

Notably, the concurrent work of Farsang et al. [82] and Danieli et al. [83] develop nonlinear SSMs and train them with DEER, with Danieli et al. [83] scaling to very strong performance as a 7B parameter language model. Both highlight the fast convergence of DEER, which is a result of the contractivity of their architectures: Farsang et al. [82] parameterizes their LrcSSM to be contractive, while Danieli et al. [83] clip the norms of their weight matrices. Ensuring a negative largest Lyapunov exponent through parameterization guarantees parallelizability for the entire training process, enabling faster and more scalable learning. Our contribution provides a theoretical foundation for why stability is essential in designing efficiently parallelizable nonlinear SSMs.

Finally, our results have implications for the *interpretation* of stable nSSMs. Because each Gauss-Newton step in DEER is a linear dynamical system (LDS), and because we prove in Theorem 4 that DEER converges in $\mathcal{O}(\log T)$ steps for a stable nSSM, we can interpret a stable nSSM as being equivalent to a "stack" of $\mathcal{O}(\log T)$ LDSs coupled by nonlinearities (cf. Appendix J).

**Limitations and Future Work**   While this work focuses on establishing the fundamental concepts and theoretical foundations, several practical considerations arise for scaling to large systems. Notably, DEER incurs a significant memory footprint. While this issue can be alleviated through quasi-Newton methods [3, 20], these approaches require more optimization steps to converge. Studying quasi-Newton methods with our theory could provide new insight into the efficacy of these methods.

Overall, the theoretical tools developed here have immediate implications for parallelizing nonlinear systems, and they open several exciting avenues for future work.

## Acknowledgments

We thank members of the Linderman Lab for helpful feedback, particularly Noah Cowan, Henry Smith, Skyler Wu, and Etaash Katiyar. We also thank Federico Danieli, Will Merrill, Julien Siems, and Riccardo Grazzi for helpful discussions. We thank the anonymous NeurIPS reviewers whose feedback improved this paper.

X.G. acknowledges support from the Walter Byers Graduate Scholarship from the NCAA. L.K. was a Goldstine Fellow at IBM Research while conducting this research. S.W.L. was supported by fellowships from the Simons Collaboration on the Global Brain, the Alfred P. Sloan Foundation, and the McKnight Foundation.

We thank Stanford University and the Stanford Research Computing Center for providing computational resources and support. Additional computations were performed on Marlowe [84], Stanford University's GPU-based Computational Instrument, supported by Stanford Data Science and Stanford Research Computing.

The authors have no competing interests to declare.

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

# Appendix

## A    Brief Overview of DEER/DeepPCR

This section provides background on DEER/DeepPCR needed to support section 4 of the main text. Other options for further background on DEER are sections 2-4 of Gonzalez et al. [3] and the corresponding blog post [85].

We begin with a brief review of DEER/DeepPCR [2, 1, 3]. As mentioned in the introduction, the choice of optimizer is crucial for this procedure to outperform sequential evaluation in terms of wall clock time. Indeed, for this reason DEER uses the Gauss-Newton method (GN) to minimize the residual loss, since GN exhibits quadratic convergence rates near the optimum [21]. Recall from eq. (3) that the $i$-th step of the DEER algorithm is,

$$\mathbf{s}^{(i+1)} = \mathbf{s}^{(i)} - \mathbf{J}(\mathbf{s}^{(i)})^{-1}\,\mathbf{r}(\mathbf{s}^{(i)}).$$

This step requires inverting the $TD \times TD$ matrix, $\mathbf{J}(\mathbf{s}^{(i)})$. Rather than explicitly inverting it, which is generally infeasible, DEER solves for the updates by running a linear time-varying recursion [3]:

$$\Delta s_t^{(i+1)} = J_t^{(i)}\,\Delta s_{t-1}^{(i+1)} - r_t(s^{(i)}), \qquad \text{where} \qquad \Delta s_t^{(i+1)} := s_t^{(i+1)} - s_t^{(i)} \qquad (13)$$

Unlike the standard sequential rollout, this recursion can be parallelized and computed in $O(\log T)$ time using a parallel associative scan [5]. When the number of optimization steps needed for DEER to converge to the true trajectory is relatively small, DEER can yield faster overall evaluation than the sequential approach. Since Gauss–Newton converges quadratically *when the initial guess is sufficiently close to the true optimum* [1, 21], DEER potentially only requires a tiny number of iterations to converge. Our first key result is to prove that DEER always converges globally with linear rate, and will thus always reach this basin of quadratic convergence after sufficient time.

**A note about notation**    The DEER quantities:

- residual $\mathbf{r}(\mathbf{s}) \in \mathbb{R}^{TD}$
- Jacobian $\mathbf{J}(\mathbf{s}) \in \mathbb{R}^{TD \times TD}$
- merit function $\mathcal{L}(\mathbf{s}) \in \mathbb{R}$

are functions of the current guess for the trajectory $\mathbf{s} = \mathrm{vec}(s_1, \ldots, s_T) \in \mathbb{R}^{TD}$. As much as possible, we try to emphasize the dependence on the current guess for the trajectory, but sometimes we will drop the dependence for notational compactness.

## B    Merit Function is PL

This section provides a proof of main text Proposition 1. We first note that Proposition 1 applies to optimizing *any* nonlinear sum of squares problem where $\mathcal{L}(\mathbf{s}) = \frac{1}{2}\|\mathbf{r}(\mathbf{s})\|_2^2$, not just the $\mathbf{r}$ we consider in this paper (defined in eq. (2)).

**Proposition** (Proposition 1)**.** *The merit function $\mathcal{L}(\mathbf{s})$ defined in eq. (2) satisfies eq. (8) for*

$$\mu := \inf_{\mathbf{s}} \sigma_{\min}^2(\mathbf{J}(\mathbf{s})).$$

*Proof.* Observe that

$$\nabla \mathcal{L}(\mathbf{s}) = \mathbf{J}(\mathbf{s})^\top \mathbf{r}(\mathbf{s}) \quad \text{and} \quad \mathcal{L}(\mathbf{s}^*) = 0.$$

Substituting these expressions into the PL inequality in eq. (8) we obtain

$$\mathbf{r}^\top \mathbf{J}(\mathbf{s})\,\mathbf{J}(\mathbf{s})^\top \mathbf{r} \ \geq \ \mu\,\mathbf{r}^\top \mathbf{r}.$$

Therefore, if $\mathbf{J}$ is full rank, then the merit function $\mathcal{L}$ is $\mu$-PL, where

$$\mu = \inf_{\mathbf{s}} \lambda_{\min}\left(\mathbf{J}(\mathbf{s})\mathbf{J}(\mathbf{s})^\top\right)$$
$$= \inf_{\mathbf{s}} \sigma_{\min}^2\left(\mathbf{J}(\mathbf{s})\right)$$

$\square$

To be precise, we must have $\mu > 0$ for $\mathcal{L}$ to satisfy the definition of PL. Therefore, a condition that must apply for $\mathcal{L}$ to be PL is that we must have $\inf_{\mathbf{s}} \sigma_{\min}(\mathbf{J}(\mathbf{s})) > 0$. We note that the proof strategy of Theorem 2 ensures that $\inf_{\mathbf{s} \in \mathbb{R}^{TD}} \sigma_{\min}(\mathbf{J}(\mathbf{s})) > 0$ if we assume eq. (22), which holds for dynamical systems that are globally contracting.

By the chain rule, eq. (22) also holds for functions of the form $f(s) = \phi(Ws)$, where $W \in \mathbb{R}^{D \times D}$ and $\phi$ is a scalar function with bounded derivative that is applied elementwise. In particular, such a function $\phi(Ws)$ satisfies eq. (22) whether or not it is globally contracting. This function class is extremely common in deep learning (nonlinearities with bounded derivatives include $\tanh$, the logistic function and $\mathrm{ReLU}$).

In our statement and proof of Proposition 1, we deliberately do not specify the set over which we take the infimum. The result is true regardless of what this set is taken to be. The largest such set would be $\mathbb{R}^{TD}$, but other sets that could be of interest are the optimization trajectory $\{\mathbf{s}^{(i)}, i \in \mathbb{N}\}$, or alternatively a neighborhood of the solution $\mathbf{s}^*$. We discuss further in Appendix C.

**Some more general notes on the PL inequality** The PL inequality or gradient dominance condition is stated differently in different texts [30, 29, 86]. We follow the presentation of Karimi et al. [28]. Karimi et al. [28] emphasizes that PL is often weaker than many other conditions that had been assumed in the literature to prove linear convergence rates.

We note that the PL inequality as stated in eq. (8) is not invariant to the scaling of $\mathcal{L}$. However, in Definition 3 of Nesterov and Polyak [30], they broaden the definition to be gradient dominant of degree $p \in [1, 2]$. The PL inequality we state in eq. (8) corresponds to gradient dominance of degree 2. Note that gradient dominance of degree 1 is scale-invariant.

## C  Merit Function PL Constant is Controlled by Largest Lyapunov Exponent of Model

This section provides the proof of main text Theorem 2.

**Theorem** (Theorem 2). *Assume that the LLE regularity condition from eq. (10) holds. Then if $\lambda \neq 0$ the PL constant $\mu$ of the merit function in (8) satisfies*

$$\frac{1}{a} \cdot \frac{e^\lambda - 1}{e^{\lambda T} - 1} \;\leq\; \sqrt{\mu} \;\leq\; \min\left(\frac{1}{b} \cdot \frac{1}{e^{\lambda(T-1)}}, 1\right). \tag{14}$$

*If $\lambda = 0$, then the bounds are instead*

$$\frac{1}{aT} \;\leq\; \sqrt{\mu} \;\leq\; \min\left(\frac{1}{b}\sqrt{\frac{2D}{T+1}}, 1\right).$$

*Proof.* We present two proofs. A shorter, direct proof of (14) assuming $\|\cdot\|$ is the standard Euclidean norm, and then a more general version in Appendix C.1, which will be useful later on.

Notice that the residual function Jacobian $\mathbf{J}$ (4) can be written as the difference of the identity and a $T$-nilpotent matrix $\mathbf{N}$, as

$$\mathbf{J} = \mathbf{I}_{TD} - \mathbf{N} \quad \text{with} \quad \mathbf{N}^T = \mathbf{0}_{TD}$$

Because $\mathbf{N}$ is nilpotent, the Neumann series for $\mathbf{J}^{-1}$ is a finite sum:

$$\mathbf{J}^{-1} = (\mathbf{I}_{TD} - \mathbf{N})^{-1} = \sum_{k=0}^{T-1} \mathbf{N}^k. \tag{15}$$

Straightforward linear algebra also shows that the norms of the powers of this nilpotent matrix are bounded, which enables one to upper bound the inverse of the Jacobian

$$\|\mathbf{N}^k\|_2 \leq a\, e^{\lambda k} \quad \text{and therefore} \quad \|\mathbf{J}^{-1}\|_2 \;\leq\; \sum_{k=0}^{T-1} \|\mathbf{N}^k\|_2 \;\leq\; \sum_{k=0}^{T-1} a\, e^{\lambda k} = a\frac{1 - e^{\lambda T}}{1 - e^\lambda}. \tag{16}$$

The powers of $\mathbf{N}$ are closely related to the dynamics of the nonlinear state space model. We provide a dynamical interpretation below, in the paragraph "The dynamical interpretation of $\mathbf{N}$ and its powers".

To lower bound $\|\mathbf{J}^{-1}\|_2$, we observe that by the SVD, a property of the spectral norm is that

$$\|\mathbf{J}^{-1}\|_2 = \sup_{\substack{\|x\|_2=1 \\ \|y\|_2=1}} x^\top \mathbf{J}^{-1} y. \tag{17}$$

We pick two unit vectors $u$ and $v$, both in $\mathbb{R}^{TD}$, that are zero everywhere other than where they need to be to pull out the bottom-left block of $\mathbf{J}^{-1}$ (i.e., the only non-zero block in $\mathbf{N}^{T-1}$, which is equal to $J_T J_{T-1} \ldots J_2$). Doing so, we get

$$u^T \mathbf{J}^{-1} v = \tilde{u}^T (J_T J_{T-1} \ldots J_2) \tilde{v},$$

where $\tilde{u}$ and $\tilde{v}$ are unit vectors in $\mathbb{R}^D$, and are equal to the nonzero entries of $u$ and $v$.

Note, therefore, that because of eq. (17), it follows that

$$\tilde{u}^T (J_T J_{T-1} \ldots J_2) \tilde{v} \leq \|\mathbf{J}^{-1}\|_2, \tag{18}$$

i.e. we also have a **lower bound** on $\|\mathbf{J}^{-1}\|_2$.

Furthermore, choosing $\tilde{u}$ and $\tilde{v}$ to make

$$\tilde{u}^T (J_T J_{T-1} \ldots J_2) \tilde{v} = \|J_T J_{T-1} \ldots J_2\|_2,$$

we can plug in this choice of $\tilde{u}$ and $\tilde{v}$ into eq. (18), to obtain

$$\|J_T J_{T-1} \ldots J_2\|_2 \leq \|\mathbf{J}^{-1}\|_2.$$

Applying the regularity conditions (10) for $k = T - 1$ and $t = 2$ we obtain

$$b \, e^{\lambda(T-1)} \leq \|\mathbf{J}^{-1}\|_2. \tag{19}$$

Because

$$\lambda_{\min}\left(\mathbf{J}\mathbf{J}^\top\right) = \frac{1}{\|\mathbf{J}^{-1}\|_2^2},$$

the result for $\lambda \neq 0$ follows by applying eq. (16) and eq. (19) at all $\mathbf{s}^{(i)}$ along the optimization trajectory.

Note that any choice of $\tilde{u}$ and $\tilde{v}$ results in a lower bound, i.e. we could also have targetted the block identity matrices. So, it also follows that $1 \leq \|\mathbf{J}^{-1}\|_2$, and so

$$\max\left(b \, e^{\lambda(T-1)}, 1\right) \leq \|\mathbf{J}^{-1}\|_2.$$

Finally, let us conclude by considering the case $\lambda = 0$. In this setting, the lower bound on $\sqrt{\mu}$ follows from L'Hôpital's rule. For the upper bound, we again must lower bound $\|\mathbf{J}^{-1}\|_2$. To do so, we leverage the relationship between spectral and Frobenius norms, namely that for an $n \times n$ matrix $A$,

$$\frac{\|A\|_F}{\sqrt{n}} \leq \|A\|_2 \leq \|A\|_F. \tag{20}$$

We can find the squared Frobenius norm, i.e. $\|\mathbf{J}^{-1}\|_F^2$, which is the sum of the squares of all of the entries. The squared Frobenius norm factors over the block structure of the matrix, i.e. $\|\mathbf{J}^{-1}\|_F^2$ is the sum of the squared Frobenius norms of the blocks. We know that each block has spectral norm lower bounded by $b$, so each block also has Frobenius norm lower bounded by $b$. Therefore, summing up over all of the blocks, it follows that

$$b^2 \frac{T(T+1)}{2} \leq \|\mathbf{J}^{-1}\|_F^2$$

and

$$\|\mathbf{J}^{-1}\|_F \leq \sqrt{TD} \|\mathbf{J}^{-1}\|_2.$$

Putting these equations together, it follows that

$$b\sqrt{\frac{T(T+1)}{2}} \leq \sqrt{TD} \|\mathbf{J}^{-1}\|_2$$

or

$$b\sqrt{\frac{T+1}{2D}} \leq \|\mathbf{J}^{-1}\|_2,$$

and so the upper bound on $\sqrt{\mu}$ when $\lambda = 0$ follows from taking reciprocals. $\square$

The above proof sheds light on how many dynamical system properties fall out of the structure of $\mathbf{J}(\mathbf{s})$, which we now discuss further.

**Discussion of why small $\sigma_{\min}(\mathbf{J}(\mathbf{s}))$ leads to ill-conditioned optimization**   Recall that our goal is to find a lower bound on the smallest singular value of $\mathbf{J}(\mathbf{s})$, which we denote by $\sigma_{\min}(\mathbf{J}(\mathbf{s}))$. This quantity controls the difficulty of optimizing $\mathcal{L}$. For example, the Gauss-Newton update is given by $\mathbf{J}(\mathbf{s})^{-1}\mathbf{r}(\mathbf{s})$. Recall that

$$\sigma_{\max}\left(\mathbf{J}(\mathbf{s})^{-1}\right) = 1/\sigma_{\min}(\mathbf{J}(\mathbf{s}))$$
$$= \|\mathbf{J}(\mathbf{s})^{-1}\|_2.$$

Recall that an interpretation of the spectral norm $\|\mathbf{J}(\mathbf{s})\|_2$ is how much multiplication by $\mathbf{J}(\mathbf{s})$ can increase the length of a vector. Therefore, we see that very small values of $\sigma_{\min}(\mathbf{J}(\mathbf{s}))$ result in large values of $\|\mathbf{J}(\mathbf{s})^{-1}\|_2$, which means that $\|\mathbf{J}(\mathbf{s})^{-1}\mathbf{r}(\mathbf{s})\|_2$ can become extremely large as well, and small perturbations in $\mathbf{r}$ can lead to very different Gauss-Newton updates (i.e. the problem is ill-conditioned, cf. Nocedal and Wright [21] Appendix A.1).

Furthermore, we observe that in the $\lambda > 0$ (unpredictable) setting and the large $T$ limit, the upper and lower bounds in (14) are tight, as they are both $\mathcal{O}(e^{\lambda(T-1)})$. Thus, the upper and lower bounds together ensure that unpredictable dynamics will suffer from degrading conditioning.

In contrast, in the $\lambda < 0$ (predictable) setting, the lower bound on $\sqrt{\mu}$ converges to $\frac{1-e^\lambda}{a}$, which is bounded away from zero and *independent of the sequence length*. Thus, in predictable dynamics, there is a lower bound on $\sigma_{\min}(\mathbf{J})$ or, equivalently, an upper bound on $\sigma_{\max}(\mathbf{J}^{-1})$.

**The dynamical interpretation of N and its powers**   As shown in the above proof,

$$\mathbf{J}(\mathbf{s})^{-1} = (\mathbf{I}_{TD} - \mathbf{N}(\mathbf{s}))^{-1} = \sum_{k=0}^{T-1} \mathbf{N}(\mathbf{s})^k.$$

It is worth noting explicitly that

$$\mathbf{N}(\mathbf{s}) = \begin{pmatrix} 0 & 0 & \dots & 0 & 0 \\ J_2 & 0 & \dots & 0 & 0 \\ \vdots & \vdots & \ddots & \vdots & \vdots \\ 0 & 0 & \dots & 0 & 0 \\ 0 & 0 & \dots & J_T & 0 \end{pmatrix} \quad \text{where} \quad J_t := \frac{\partial f_t}{\partial s_{t-1}}(s_{t-1}), \tag{21}$$

i.e. $\mathbf{N}(\mathbf{s})$ collects the Jacobians of the dynamics function along the first lower diagonal. Each matrix power $\mathbf{N}^k$ therefore collects length $k$ products along the $k$th lower diagonal. Thus, multiplication by $\mathbf{J}(\mathbf{s})^{-1} = \sum_{k=0}^{T-1} \mathbf{N}(\mathbf{s})^k$ recovers running forward a linearized form of the dynamics, which is one of the core insights of DeepPCR and DEER [2, 1].

Concretely, in the setting where $T = 4$, we have

$$\mathbf{N}^0 = \begin{pmatrix} I_D & 0 & 0 & 0 \\ 0 & I_D & 0 & 0 \\ 0 & 0 & I_D & 0 \\ 0 & 0 & 0 & I_D \end{pmatrix}$$

$$\mathbf{N} = \begin{pmatrix} 0 & 0 & 0 & 0 \\ J_2 & 0 & 0 & 0 \\ 0 & J_3 & 0 & 0 \\ 0 & 0 & J_4 & 0 \end{pmatrix}$$

$$\mathbf{N}^2 = \begin{pmatrix} 0 & 0 & 0 & 0 \\ 0 & 0 & 0 & 0 \\ J_3 J_2 & 0 & 0 & 0 \\ 0 & J_4 J_3 & 0 & 0 \end{pmatrix}$$

$$\mathbf{N}^3 = \begin{pmatrix} 0 & 0 & 0 & 0 \\ 0 & 0 & 0 & 0 \\ 0 & 0 & 0 & 0 \\ J_4 J_3 J_2 & 0 & 0 & 0 \end{pmatrix}$$

$$\mathbf{J}^{-1} = \begin{pmatrix} I_D & 0 & 0 & 0 \\ J_2 & I_D & 0 & 0 \\ J_3 J_2 & J_3 & I_D & 0 \\ J_4 J_3 J_2 & J_4 J_3 & J_4 & I_D \end{pmatrix}$$

**Connection to semiseparable matrices and Mamba2**  Having depicted the structure of $\mathbf{J}^{-1}$, we note the connection between $\mathbf{J}^{-1}$ in this paper and the attention or sequence mixer matrix $M$ in Dao and Gu [87], which introduced the Mamba2 architecture (see equation 6 or Figure 2 of Dao and Gu [87] for the form of $M$, and compare with $\mathbf{J}^{-1}$ above).

Mamba2 is a deep learning sequence modeling architecture. Its sequence mixer in each layer has at its core a linear dynamical system. Dao and Gu [87] observe that while a linear dynamical system (LDS) can be evaluated recurrently (sequentially) or in parallel (for example, with a parallel scan), it can also be evaluated multiplying the inputs to the LDS by the matrix $M$. Since each DEER iteration is also a linear dynamical system, with the transition matrices given by $\{J_t\}_{t=2}^T$, it follows that $M$ in Dao and Gu [87] and $\mathbf{J}^{-1}$ in our paper are the same object, and so results about these objects from these two papers transfer.

In particular, we observe that, in the language from Dao and Gu [87], the $\mathbf{J}^{-1}$ we consider in this paper is $D$-*semiseparable* (see Definition 3.1 from Dao and Gu [87]). Thus, any efficient, hardware-aware algorithms and implementations developed for $D$-semiseparable matrices could also be applied to accelerating each iteration of DEER, though we note that Dao and Gu [87] focus on the 1-semiseparable setting, which they call a *state space dual* or *SSD* layer. In any case, using these connections to accelerate each iteration of DEER and related parallel Newton algorithms from a systems implementation perspective would be an interesting direction for future work.

**A framing of Theorem 2 based on global bounds on $\|J_t\|_2$**  We chose to prove Theorem 2 using condition (10) in order to highlight the natural connection between the smallest singular value of $\mathbf{J}$ and system stability (as measured by its LLE). However, an assumption with a different framing would be to impose a uniform bound on the spectral norm of the Jacobian over the entire state space:

$$\sup_{s \in \mathbb{R}^D} \|J(s)\|_2 \leq \rho. \tag{22}$$

For $\rho < 1$, this assumption corresponds to global contraction of the dynamics [7].

If we replace the LLE regularity condition (10) with the global spectral norm bound (22) in the proof of Theorem 2, we obtain that the PL constant is bounded away from zero, i.e.

$$\frac{1}{a} \cdot \frac{\rho - 1}{\rho^T - 1} \leq \sqrt{\inf_{\mathbf{s} \in \mathbb{R}^{TD}} \sigma_{\min}^2(\mathbf{J}(\mathbf{s}))}.$$

In particular, if the dynamics are contracting everywhere (i.e., $\rho < 1$), the condition (22) guarantees good conditioning of $\mathbf{J}$ throughout the entire state space.

**Discussion of the LLE regularity conditions** The LLE regularity conditions in eq. (10) highlight the more natural "average case" behavior experienced along actual trajectories $\mathbf{s} \in \mathbb{R}^{TD}$. This "average case" behavior is highlighted, for example, by our experiments with the two-well system (cf. Section 5 and Appendix K.4), where even though a global upper bound on $\|J_t(s_t)\|_2$ over all of state space would be greater than 1 (i.e., there are unstable regions of state space), we observe fast convergence of DEER because the system as a whole has negative LLE (its trajectories are stable on average).

We also note the pleasing relationship the LLE regularity conditions have with the definition of the LLE given in eq. (5). Note that in the LLE regularity conditions in eq. (10), the variable $k$ denotes the sequence length under consideration. Taking logs and dividing by $k$, we therefore obtain

$$\frac{\log b}{k} + \lambda \le \frac{1}{k}\log\left(\|J_{t+k-1}J_{t+k-2}\cdots J_t\|\right) \le \frac{\log a}{k} + \lambda.$$

Therefore, as $k \to T$, and as $T \to \infty$ (i.e., we consider longer and longer sequences), we observe that the finite-time estimates of the LLE converge to the true LLE $\lambda$.

We observe that as $\mathbf{s}^{(i)}$ approaches the true solution $\mathbf{s}^*$, the regularity conditions in eq. (10) become increasingly reasonable. Since any successful optimization trajectory must eventually enter a neighborhood of $\mathbf{s}^*$, it is natural to expect these conditions to hold there. In fact, rather than requiring the regularity conditions over all of state space or along the entire optimization trajectory, one could alternatively assume that they hold within a neighborhood of $\mathbf{s}^*$, and prove a corresponding version of Theorem 2.

We now do so, using the additional assumption that $\mathbf{J}$ is $L$-Lipschitz.

**Theorem 6.** *If $\mathbf{J}$ is $L$-Lipschitz, then there exists a ball of radius $R$ around the solution $\mathbf{s}^*$, denoted $B(\mathbf{s}^*, R)$, such that*

$$\forall \mathbf{s} \in B(\mathbf{s}^*, R) \qquad |\sigma_{\min}(\mathbf{J}(\mathbf{s})) - \sigma_{\min}(\mathbf{J}(\mathbf{s}^*))| \le LR$$

*Proof.* The argument parallels the proof of Theorem 2 in Liu et al. [88].

A fact stemming from the reverse triangle inequality is that for any two matrices $\mathbf{A}$ and $\mathbf{B}$,

$$\sigma_{\min}(\mathbf{A}) \ge \sigma_{\min}(\mathbf{B}) - \|\mathbf{A} - \mathbf{B}\|.$$

Applying this with $\mathbf{A} = \mathbf{J}(\mathbf{s})$ and $\mathbf{B} = \mathbf{J}(\mathbf{s}^*)$, we obtain

$$\sigma_{\min}(\mathbf{J}(\mathbf{s})) \ge \sigma_{\min}(\mathbf{J}(\mathbf{s}^*)) - \|\mathbf{J}(\mathbf{s}) - \mathbf{J}(\mathbf{s}^*)\|.$$

If the Jacobian $\mathbf{J}(\cdot)$ is $L$-Lipschitz, then

$$\|\mathbf{J}(\mathbf{s}) - \mathbf{J}(\mathbf{s}^*)\| \le L\|\mathbf{s} - \mathbf{s}^*\|.$$

Combining, we get

$$\sigma_{\min}(\mathbf{J}(\mathbf{s})) \ge \sigma_{\min}(\mathbf{J}(\mathbf{s}^*)) - L\|\mathbf{s} - \mathbf{s}^*\|$$

and

$$\sigma_{\min}(\mathbf{J}(\mathbf{s}^*)) \ge \sigma_{\min}(\mathbf{J}(\mathbf{s})) - L\|\mathbf{s} - \mathbf{s}^*\|,$$

which gives

$$\sigma_{\min}(\mathbf{J}(\mathbf{s}^*)) - L\|\mathbf{s} - \mathbf{s}^*\| \le \sigma_{\min}(\mathbf{J}(\mathbf{s})) \le \sigma_{\min}(\mathbf{J}(\mathbf{s}^*)) + L\|\mathbf{s} - \mathbf{s}^*\|.$$

Ensuring that $\|\mathbf{s} - \mathbf{s}^*\| \le R$ completes the proof. $\square$

A consequence of Theorem 6 is that if the system is unpredictable, then there exists a finite ball around $\mathbf{s}^*$ where the conditioning of the merit function landscape is provably bad.

As a concrete example, suppose that $\sigma_{\min}(\mathbf{J}(\mathbf{s}^*)) = \epsilon$ and $L = 1$. Then *at best*, the PL constant of the loss function inside the ball $B(\mathbf{s}^*, R)$ is $\epsilon + R$. If $\epsilon$ is small (bad conditioning) then $R$ can be chosen such that the PL constant inside the ball $B(\mathbf{s}^*, R)$ is also small.

**Controlling $\sigma_{\max}(\mathbf{J})$**  In our proof of Theorem 2, we proved upper and lower bounds for $\sigma_{\min}(\mathbf{J}(\mathbf{s}))$ that depended on the sequence length $T$. We can also prove upper and lower bounds for $\sigma_{\max}(\mathbf{J}(\mathbf{s}))$, but these do not depend on the sequence length.

Assuming condition (22), an upper bound on $\sigma_{\max}(\mathbf{J})$ is straightforward to compute via the triangle inequality,

$$\begin{aligned} \sigma_{\max}(\mathbf{J}) &= \|\mathbf{J}\|_2 \\ &= \|\mathbf{I} - \mathbf{N}\|_2 \\ &\leq 1 + \|\mathbf{N}\|_2. \end{aligned}$$

Recalling the definition of $\mathbf{N}$ in (21), we observe that it is composed of $\{J_t\}$ along its lower block diagonal, and so we have

$$\|\mathbf{N}(\mathbf{s})\|_2 = \sup_t \|J_t(s_t)\|$$

$$\sup_{\mathbf{s}\in\mathbb{R}^{TD}} \|\mathbf{N}(\mathbf{s})\|_2 = \sup_{s\in\mathbb{R}^D} \|J(s)\|$$

Elaborating, for a particular choice of trajectory $\mathbf{s} \in \mathbb{R}^{TD}$, $\|\mathbf{N}(\mathbf{s})\|_2$ is controlled by the maximum spectral norm of the Jacobians $J_t(s_t)$ along this trajectory. Analogously, $\sup_{\mathbf{s}\in\mathbb{R}^{TD}} \|\mathbf{N}(\mathbf{s})\|_2$— i.e., the supremum of the spectral norm of $\mathbf{N}(\mathbf{s})$ over all possible trajectories $\mathbf{s} \in \mathbb{R}^{TD}$, i.e. the optimization space—is upper bounded by $\sup_{s\in\mathbb{R}^D} \|J(s)\|_2$, i.e. the supremum of the spectral norm of the system Jacobians over the state space $\mathbb{R}^D$.

Thus, it follows that

$$\sigma_{\max}(\mathbf{J}) \leq 1 + \rho. \tag{23}$$

Importantly, the upper bound on $\sigma_{\max}(\mathbf{J})$ does not scale with the sequence length $T$.

To obtain the lower bound on $\sigma_{\max}(\mathbf{J})$, we notice that it has all ones along its main diagonal, and so simply by using the unit vector $\mathbf{e}_1$, we obtain

$$\mathbf{e}_1^\top \mathbf{J} \mathbf{e}_1 = 1 \leq \sigma_{\max}(\mathbf{J}). \tag{24}$$

**Condition number of J**  Note that the condition number $\kappa$ of a matrix is defined as the ratio of its maximum and minimum singular values, i.e.

$$\kappa(\mathbf{J}) = \frac{\sigma_{\max}(\mathbf{J})}{\sigma_{\min}(\mathbf{J})}.$$

However, because our bounds in eq. (23) and eq. (24) on $\sigma_{\max}(\mathbf{J})$ do not scale with the sequence length $T$, it follows that the scaling with $T$ of an upper bound on $\kappa(\mathbf{J})$—the conditioning of the optimization problem—is controlled solely by the bounds on $\sigma_{\min}(\mathbf{J})$ that we provided in Theorem 2. The importance of studying how the conditioning scales with $T$ stems from the fact that we would like to understand if there are regimes—particularly involving large sequence lengths and parallel computers—where parallel evaluation can be faster than sequential evaluation.

## C.1  A Generalized Proof that the Largest Lyapunov Exponent Controls the PL Constant

**Lower Singular Value Bound**  Recall the following sequence of observations.

$$\lambda_{\min}(\mathbf{J}\mathbf{J}^\top) = \sigma_{\min}^2(\mathbf{J}) = \frac{1}{\sigma_{\max}^2(\mathbf{J}^{-1})} = \frac{1}{\|\mathbf{J}^{-1}\|_2^2}$$

Thus, to lower bound the eigenvalues of $\mathbf{J}\mathbf{J}^\top$ as desired, we can *upper bound* the spectral norm of $\mathbf{J}^{-1}$.

**General Bound**  As discussed in the main text, the predictability of the nonlinear state space model is characterized by the products of its Jacobians along a trajectory. We will need to control how this product behaves. To reduce notational burden, we will drop the DEER iteration superscript $i$. In particular, we will assume that there exists a function $g_J : \mathbb{N}_0 \to \mathbb{R}$ such that

$$\big\| J_{k-1} J_{k-2} \cdots J_i \big\|_\xi \leq g_J(k - i)$$

holds for all products $J_{k-1} \cdots J_i$ with $k > i$, where $\| \cdot \|_\xi$ is the matrix operator norm induced by the vector norm $\| \cdot \|_\xi$. Intuitively, the function $g_J$ measures the stability of the nonlinear state space model. For example, suppose the model is contracting with rate $\rho < 1$. Then the product of Jacobians exponentially decreases, which we can write as

$$g_J(j) = a \, \rho^j,$$

for some $a \geq 1$. The larger the value of $a$, the larger the potential "overshoot", before exponential shrinkage begins.

**Lemma 1.** Let $\| \cdot \|_\xi$ be the matrix operator norm induced by the vector norm $\| \cdot \|_\xi$. Suppose there is a function $g_J : \mathbb{N}_0 \to \mathbb{R}$ such that

$$\left\| J_{k-1} J_{k-2} \cdots J_i \right\|_\xi \leq g_J(k - i)$$

holds for all products $J_{k-1} \cdots J_i$ with $k > i$. Define

$$G_J(T) = \sum_{0 \leq j < T} g_J(j).$$

Then

$$\|\mathbf{J}^{-1}\|_\xi \leq G_J(T).$$

*Proof.* Let $\mathbf{y} = \mathbf{J}^{-1}\mathbf{x}$. By backward substitution for the blockwise entries of $\mathbf{J}^{-1}\mathbf{x}$, we have

$$y_k = \sum_{i \in [k]} \left( J_{k-1} J_{k-2} \cdots J_i \right) x_i.$$

Omitting the subscript $\xi$ in the norms for brevity and applying the triangle inequality and the induced-norm property,

$$\|\mathbf{y}\| \leq \sum_{k \in [T]} \|y_k\| \leq \sum_{k \in [T]} \sum_{i \in [k]} \|J_{k-1} \cdots J_i\| \, \|x_i\|.$$

By assumption, $\|J_{k-1} \cdots J_i\| \leq g_J(k - i)$. Hence,

$$\|\mathbf{y}\| \leq \sum_{k \in [T]} \sum_{i \in [k]} g_J(k - i) \|x_i\| = \sum_{i \in [T]} \|x_i\| \sum_{k=i}^{T} g_J(k - i) = \sum_{i \in [T]} \|x_i\| \, G_J\big(T - i + 1\big).$$

Since $G_J(t)$ is nondecreasing in $t$, the largest multiplier in these sums is $G_J(T)$. In the worst case, $\|\mathbf{x}\| = \|x_1\|$. Thus,

$$\|\mathbf{J}^{-1}\| \leq \frac{\|\mathbf{J}^{-1}\mathbf{x}\|}{\|\mathbf{x}\|} = \frac{\|\mathbf{y}\|}{\|\mathbf{x}\|} \leq G_J(T).$$

This completes the proof. $\qquad \square$

**Remark 1** (Contraction in The Identity Metric). Recall that a system is contracting in the identity metric when the system Jacobians have singular values less than one:

$$\forall i, \quad \|J_i\| \leq \rho \quad \Longleftrightarrow \quad J_i^\top J_i \leq \rho^2 I$$

In this case, we can take

$$g_J(j) = \rho^j.$$

Then, by Lemma 1,

$$\|\mathbf{J}^{-1}\| \leq \sum_{j=0}^{T-1} \rho^j = \begin{cases} \dfrac{\rho^T - 1}{\rho - 1}, & \rho \neq 1, \\ T, & \rho = 1, \end{cases} \tag{25}$$

where in the case $\rho = 1$, there are $T$ summands and each term equals 1.

**Remark 2** (Contraction in Time-Varying, State-Dependent Metrics). Recall that a system is contracting in metric $M_i = M(s_i, i)$ if the following linear matrix inequality is satisfied

$$\forall i \in [T-1], \quad J_i^\top M_{i+1} J_i \preceq e^{2\lambda} M_i.$$

Equivalently, this condition can be written as a norm constraint

$$\forall i \in [T-1], \quad \|M_{i+1}^{1/2} J_i M_i^{-1/2}\| \leq \rho$$

Using these metrics, we define the block-diagonal, symmetric, positive-definite matrix

$$\mathbf{M} = \mathrm{diag}(M_1, M_2, \ldots, M_T)$$

as well as the similarity transform of the residual function Jacobian, based on this matrix

$$\mathbf{J_M} := \mathbf{M}^{1/2} \, \mathbf{J} \, \mathbf{M}^{-1/2}.$$

Then the off-diagonal block entries of $\mathbf{J_M}$ are

$$M_{i+1}^{1/2} J_i M_i^{-1/2} \quad \text{for} \quad i \in [T-1],$$

while its diagonal block entries are the identity matrix. If the off-diagonal blocks of $J_M$ satisfy a product bound function $g_{\mathbf{J_M}}(j)$ as in Lemma 1, then $\mathbf{J_M}$ has norm bounded by $G_{\mathbf{J_M}}(T)$. Hence,

$$\|\mathbf{J}^{-1}\| = \left\|\mathbf{M}^{-1/2} \, \mathbf{M}^{1/2} \mathbf{J}^{-1} \mathbf{M}^{-1/2} \, \mathbf{M}^{1/2}\right\|$$

$$\leq \|\mathbf{M}^{-1/2}\| \, \left\|\mathbf{M}^{1/2} \mathbf{J}^{-1} \mathbf{M}^{-1/2}\right\| \, \|\mathbf{M}^{1/2}\|$$

$$= \|\mathbf{M}^{-1/2}\| \, \|\mathbf{J_M}^{-1}\| \, \|\mathbf{M}^{1/2}\|$$

$$\leq \|\mathbf{M}^{-1/2}\| \, G_{\mathbf{J_M}}(T) \, \|\mathbf{M}^{1/2}\|$$

$$= \kappa_M \, G_{\mathbf{J_M}}(T),$$

where

$$\kappa_M := \sqrt{\frac{\lambda_{\max}(\mathbf{M})}{\lambda_{\min}(\mathbf{M})}}.$$

In this case, we may again take $g_{J_M}(j) = \rho^j$, and we obtain the bound

$$\|\mathbf{J}^{-1}\| \leq \kappa_M \sum_{0 \leq j < T} \rho^j = \begin{cases} \kappa_M \dfrac{\rho^T - 1}{\rho - 1}, & \rho \neq 1, \\ \kappa_M T, & \rho = 1, \end{cases}.$$

**Remark 3** (Contraction After Burn-In). Suppose that

$$g_J(j) \leq a e^{-\lambda j}$$

where $a \geq 1$ and measures the degree of "overshoot" the system can undergo before eventually converging, and $\lambda > 0$. In particular, assume for concreteness that

$$\|J_t\| \leq 1$$

Then, the product of two Jacobians can *grow*, if $a > e^\lambda$, since

$$\|J_{t+1} J_t\| \leq a e^{-\lambda}.$$

In general, the product of Jacobians can transiently grow (i.e., overshoot) for

$$k_{\text{overshoot}} = \frac{1}{\lambda} \log a$$

time steps, at which point the product of $k > k_{\text{overshoot}}$ Jacobians will remain less than 1, and will in fact decay to zero exponentially with rate $\lambda$.

In this case, by Lemma 1:

$$\|\mathbf{J}^{-1}\| \leq a \sum_{j=0}^{T-1} e^{-\lambda j} = a \, \frac{e^{-\lambda T} - 1}{e^{-\lambda} - 1}.$$

# D  DEER Merit Function Inherits Lipschitzness of Dynamics Jacobians

This section provides a proof of main text Theorem 3.

**Theorem** (Theorem 3). *If the dynamics of the underlying nonlinear state space model have L-Lipschitz Jacobians, i.e.,*

$$\forall t > 1, \quad s, s' \in \mathbb{R}^D : \quad \|J_t(s) - J_t(s')\| \leq L\|s - s'\|,$$

*then the residual function Jacobian $\mathbf{J}$ is also L-Lipschitz, with the same L.*

*Proof.* By assumption, for each $t$,

$$\forall s, s' \in \mathbb{R}^D : \quad \|J_t(s_t) - J_t(s'_t)\|_2 \ \leq \ L\,\|s_t - s'_t\|_2.$$

Define $D_t := J_t(s'_t) - J_t(s_t)$ and

$$\mathbf{D} \ := \ \mathbf{J}(\mathbf{s}') - \mathbf{J}(\mathbf{s}).$$

Since $\mathbf{D}$ places the blocks $D_t$ along one subdiagonal, we have

$$\|\mathbf{D}\|_2 \ = \ \max_t \|D_t\|_2.$$

But each block $D_t$ satisfies the Lipschitz bound

$$\|D_t\|_2 \ \leq \ L\,\|s'_t - s_t\|_2,$$

so

$$\|\mathbf{D}\|_2 \ = \ \max_t \|D_t\|_2 \ \leq \ L\,\max_t \|s'_t - s_t\|_2 \ \leq \ L\,\|\mathbf{s}' - \mathbf{s}\|_2.$$

Hence, it follows that

$$\|\mathbf{J}(\mathbf{s}') - \mathbf{J}(\mathbf{s})\|_2 \ = \ \|\mathbf{D}\|_2 \ \leq \ L\,\|\mathbf{s}' - \mathbf{s}\|_2.$$

Thus $\mathbf{J}$ is $L$-Lipschitz. $\qquad\square$

# E  DEER Always Converges Linearly

This section provides a proof of Theorem 4.

While proofs of global convergence are challenging in general for GN, DEER is highly structured, and this can be exploited to provide a global proof of convergence. In particular, we will exploit the *hierarchical* nature of DEER, which is reflected in the fact that $\mathbf{J}$ and $\mathbf{J}^{-1}$ are lower block-triangular.

**Theorem** (Theorem 4). *Let the DEER (Gauss–Newton) updates be given by eq. (3), and let $\mathbf{s}^{(i)}$ denote the $i$-th iterate. Let $\mathbf{e}^{(i)} := \mathbf{s}^{(i)} - \mathbf{s}^*$ denote the error at iteration $i$, and assume the regularity condition in eq. (10). Then the error converges to zero at a linear rate:*

$$\|\mathbf{e}^{(i)}\|_2 \leq \chi_w\,\beta^i\|\mathbf{e}^{(0)}\|_2,$$

*for some constant $\chi_w \geq 1$ independent of $i$, and a convergence rate $0 < \beta < 1$.*

*Proof.* Our general strategy for deriving DEER convergence bounds will be to fix some weighted norm $\|\cdot\|_W := \|\mathbf{W}^{1/2} \cdot \mathbf{W}^{-1/2}\|_2$ such that each DEER step is a contraction, with contraction factor $\beta \in [0, 1)$. This will imply that the DEER error iterates decay to zero with linear rate, as

$$\|\mathbf{e}^{(i)}\|_W \leq \beta^i\|\mathbf{e}^{(0)}\|_W.$$

To convert this bound back to standard Euclidean space, we incur an additional multiplicative factor that depends on the conditioning of $\mathbf{W}$:

$$\|\mathbf{e}^{(i)}\|_2 \leq \chi_w\,\beta^i\|\mathbf{e}^{(0)}\|_2, \qquad \text{where} \qquad \chi_w := \sqrt{\frac{\lambda_{\max}(\mathbf{W})}{\lambda_{\min}(\mathbf{W})}}. \tag{26}$$

**DEER as a Contraction Mapping**   Recall that the DEER (Gauss-Newton) updates are given by

$$\mathbf{s}^{(i+1)} = \mathbf{s}^{(i)} - \mathbf{J}^{-1}(\mathbf{s}^{(i)})\mathbf{r}(\mathbf{s}^{(i)})$$

Recalling that $\mathbf{r}(\mathbf{s}^*) = \mathbf{0}$ and subtracting the fixed point $\mathbf{s}^*$ from both sides, we have that

$$\mathbf{e}^{(i+1)} = \mathbf{e}^{(i)} - \mathbf{J}^{-1}(\mathbf{s}^{(i)})\mathbf{r}^{(i)} + \mathbf{J}^{-1}(\mathbf{s}^{(i)})\,\mathbf{r}(\mathbf{s}^*) = \mathbf{e}^{(i)} - \mathbf{J}^{-1}(\mathbf{s}^{(i)})\left(\mathbf{r}(\mathbf{s}^{(i)}) - \mathbf{r}(\mathbf{s}^*)\right).$$

This equation can be written using the mean value theorem as

$$\mathbf{e}^{(i+1)} = \left(\mathbf{I} - \mathbf{J}^{-1}(\mathbf{s}^{(i)})\mathbf{B}^{(i)}\right)\mathbf{e}^{(i)} \qquad \text{where} \qquad \mathbf{B}^{(i)} := \int_0^1 \mathbf{J}(\mathbf{s}^* + \tau\mathbf{e}^{(i)})\,d\tau$$

From this, we can conclude that the DEER iterates will converge (i.e., the error shrinks to zero) if

$$\|\mathbf{I} - \mathbf{J}^{-1}(\mathbf{s}^{(i)})\mathbf{B}^{(i)}\|_W = \|\mathbf{J}^{-1}(\mathbf{s}^{(i)})\left(\mathbf{J}(\mathbf{s}^{(i)}) - \mathbf{B}^{(i)}\right)\|_W \le \beta < 1. \tag{27}$$

**Constructing the Weighted Norm**   We will choose a diagonal weighted norm, given by

$$\mathbf{W} := \mathrm{Diag}\left(I_D,\ w^2 I_D,\ \ldots,\ w^{2T} I_D\right) \in \mathbb{R}^{TD \times TD}, \qquad w > 0. \tag{28}$$

Under the norm induced by (28) we have

$$\|\mathbf{J}(\mathbf{s}^{(i)}) - \mathbf{B}^{(i)}\|_W \le 2w\rho, \tag{29}$$

$$\|\mathbf{J}^{-1}(\mathbf{s}^{(i)})\|_W \le a\frac{1 - (we^\lambda)^T}{1 - we^\lambda}, \tag{30}$$

where $\rho$ upper bounds $\|J\|_2$ over all states in the DEER optimization trajectory.

Multiplying (29) and (30) yields

$$\|\mathbf{J}^{-1}(\mathbf{s}^{(i)})\|_W \|\mathbf{J}(\mathbf{s}^{(i)}) - \mathbf{B}^{(i)}\|_W \le 2aw\rho\,\frac{1 - (we^\lambda)^T}{1 - we^\lambda}. \tag{31}$$

To ensure the right-hand side of (31) does not exceed a prescribed $\beta \in [0, 1)$, choose

$$w = \frac{\beta}{2\rho a + \beta e^\lambda}. \tag{32}$$

With this choice,

$$we^\lambda < 1, \qquad \text{and} \qquad \frac{2aw\rho}{1 - we^\lambda} = \beta, \tag{33}$$

so the geometric series in (30) is convergent and the bound in (31) holds for all $T$, because

$$\|\mathbf{J}^{-1}(\mathbf{s}^{(i)})\|_W \|\mathbf{J}(\mathbf{s}^{(i)}) - \mathbf{B}^{(i)}\|_W \le 2aw\rho\,\frac{1 - (we^\lambda)^T}{1 - we^\lambda} = \beta\left(1 - (we^\lambda)^T\right) \le \beta.$$

This shows that we can always pick a weighted norm so that DEER converges with linear rate *in that norm*. Converting back into the standard Euclidean norm using (26) and substituting in the condition number of $\mathbf{W}^{1/2}$ one finds that

$$\|\mathbf{e}^{(i)}\|_2 \le \left(\frac{2\rho a + \beta e^\lambda}{\beta}\right)^T \beta^i \|\mathbf{e}^{(0)}\|_2. \tag{34}$$

Thus, the DEER error converges with linear rate towards zero. □

**Remark 4.** The multiplicative overshoot factor arising from the conditioning of $\mathbf{W}$ grows exponentially in the sequence length $T$, leading potentially to long convergence times. Indeed, a quick calculation shows that the number of steps needed to bring the DEER error to $\epsilon$ is upper bounded as $O(T)$ because of this multiplicative constant.

**Remark 5.** One can ask under what conditions choosing $w = 1$ in (32) is possible, which eliminates the overshoot. We will address this in more detail in the next section. To provide a simple result here, we can assume that the system is contracting at every time step so that

$$\rho = e^\lambda,$$

and $a = 1$. Then we have that

$$1 = \frac{\beta}{2\rho a + \beta e^\lambda} = \frac{\beta}{2e^\lambda + \beta e^\lambda}.$$

Solving for $\lambda$, we have that if

$$\lambda \leq \log\left(\frac{\beta}{2+\beta}\right) < -\log(3),$$

then $w$ can be chosen to be equal to one, meaning the DEER converges globally with rate $\beta$ and no overshoot.

## F  DEER Converges Globally with Small Overshoot for Sufficiently Strongly Contracting Systems

In this section we show that DEER converges globally to the optimum $\mathbf{s}^*$ when the nonlinear state space model (1) is sufficiently strongly contracting. To do so, we first briefly recall the assumptions of Lemma 1. Let $\|\cdot\|_\xi$ be the matrix operator norm induced by the vector norm $\|\cdot\|_\xi$. Suppose there is a function $g_J : \mathbb{N}_0 \to \mathbb{R}$ such that

$$\left\| J_{k-1} J_{k-2} \cdots J_i \right\|_\xi \leq g_J(k-i)$$

holds for all products $J_{k-1} \cdots J_i$ with $k > i$. Define

$$G_J(T) = \sum_{0 \leq j < T} g_J(j).$$

Then

$$\|\mathbf{J}^{-1}\|_\xi \leq G_J(T).$$

For example, if there is no structure which can be exploited in the products of Jacobians $J_t$, we may consider the "one-step" growth/decay factor

$$\forall t, \quad \|J_t\| \leq e^\lambda,$$

which yields

$$g_J(j) = e^{\lambda j} \qquad \Longrightarrow \qquad G_J(T) = \sum_{0 \leq j < T} g_J(j) = \frac{1 - e^{\lambda T}}{1 - e^\lambda}.$$

**Theorem.** *DEER exhibits linear, global convergence to the optimum $\mathbf{s}^*$ with rate $\beta \in [0, 1)$ in the matrix operator norm $\|\cdot\|_\xi$ if*

$$2g_J(1)G_J(T) \leq \beta$$

*Proof.* Recall that the DEER (Gauss-Newton) updates are given by

$$\mathbf{s}^{(i+1)} = \mathbf{s}^{(i)} - \mathbf{J}^{-1}(\mathbf{s}^{(i)})\mathbf{r}(\mathbf{s}^{(i)})$$

Define the error at DEER iteration $(i)$ as $\mathbf{e}^{(i)} = \mathbf{s}^{(i)} - \mathbf{s}^*$. Recalling that $\mathbf{r}(\mathbf{s}^*) = \mathbf{0}$ and subtracting the fixed point $\mathbf{s}^*$ from both sides, we have that

$$\mathbf{e}^{(i+1)} = \mathbf{e}^{(i)} - \mathbf{J}^{-1}(\mathbf{s}^{(i)})\mathbf{r}^{(i)} + \mathbf{J}^{-1}(\mathbf{s}^{(i)})\,\mathbf{r}(\mathbf{s}^*) = \mathbf{e}^{(i)} - \mathbf{J}^{-1}(\mathbf{s}^{(i)})\left(\mathbf{r}(\mathbf{s}^{(i)}) - \mathbf{r}(\mathbf{s}^*)\right).$$

This equation can be written in terms of the mean value theorem as

$$\mathbf{e}^{(i+1)} = \left(\mathbf{I} - \mathbf{J}^{-1}(\mathbf{s}^{(i)})\mathbf{B}^{(i)}\right)\mathbf{e}^{(i)} \qquad \text{where} \qquad \mathbf{B}^{(i)} := \int_0^1 \mathbf{J}(\mathbf{s}^* + \tau\mathbf{e}^{(i)})\,d\tau$$

This follows from the identity:

$$\mathbf{r}(\mathbf{s}^{(i)}) - \mathbf{r}(\mathbf{s}^*) := \int_0^1 \mathbf{J}(\tau \mathbf{s}^{(i)} + (1-\tau)\mathbf{s}^*)\, d\tau\, (\mathbf{s} - \mathbf{s}^*) = \left( \int_0^1 \mathbf{J}(\mathbf{s}^* + \tau \mathbf{e}^{(i)}))\, d\tau \right) \mathbf{e}^{(i)}$$

This identity can be proven by starting from the fundamental theorem of calculus, by letting

$$\mathbf{s}(\tau) = \mathbf{s}^* + \tau \mathbf{e}^{(i)} = \mathbf{s}^* + \tau(\mathbf{s}^{(i)} - \mathbf{s}^*), \quad \tau \in [0, 1],$$

which defines a straight-line path from $\mathbf{s}^*$ to $\mathbf{s}^{(i)}$. The fundamental theorem of calculus then says that

$$\mathbf{r}(\mathbf{s}^{(i)}) - \mathbf{r}(\mathbf{s}^*) = \int_0^1 \frac{d}{d\tau}\mathbf{r}(\mathbf{s}(\tau))\, d\tau.$$

Applying the chain rule inside the integral gives the result, because

$$\frac{d}{d\tau}\mathbf{r}(\mathbf{s}(\tau)) = \mathbf{J}(\mathbf{s}(\tau)) \cdot \frac{d}{d\tau}\mathbf{s}(\tau) = \mathbf{J}(\mathbf{s}^* + \tau \mathbf{e}^{(i)}) \cdot \mathbf{e}^{(i)}.$$

From this, we can conclude that the DEER iterates will converge (i.e., the error shrinks to zero) if

$$\|\mathbf{I} - \mathbf{J}^{-1}(\mathbf{s}^{(i)})\mathbf{B}^{(i)}\|_\xi = \|\mathbf{J}^{-1}(\mathbf{s}^{(i)}) \left( \mathbf{J}(\mathbf{s}^{(i)}) - \mathbf{B}^{(i)} \right) \|_\xi \leq \beta < 1. \tag{35}$$

By Lemma 1 we have that

$$\|\mathbf{e}^{(i+1)}\|_\xi \leq \|\mathbf{J}^{-1}(\mathbf{s}^{(i)})\|_\xi \|\mathbf{J}(\mathbf{s}^{(i)}) - \mathbf{B}^{(i)}\|_\xi \|\mathbf{e}^{(i)}\|_\xi \leq 2\, g_J(1) G_J(T)\, \|\mathbf{e}^{(i)}\|.$$

Thus, if there exists some $\beta \in [0, 1)$ such that

$$2\, g_J(1) G_J(T) \leq \beta,$$

then the DEER error converges globally to zero in the weighted norm:

$$\|\mathbf{e}^{(i)}\|_\xi \leq \beta^i \|\mathbf{e}^{(0)}\|_\xi.$$

$\square$

**Corollary.** *Suppose the state space model is contracting in constant metric $M$, i.e.,*

$$\|M^{1/2} J_t M^{-1/2}\| = \|J_t\|_M \leq e^\lambda < 1.$$

*If $e^\lambda$ is sufficiently small, in particular if*

$$e^\lambda \leq \frac{\beta}{2 + \beta} < \frac{1}{3},$$

*then the DEER errors converge to zero with rate $\beta$.*

*Proof.* Suppose the state space model is contracting in constant metric $M$, so that

$$\|M^{1/2} J_t M^{-1/2}\| = \|J_t\|_M \leq e^\lambda < 1$$

for all $t$. Then, by Lemma 1 we have that

$$\|\mathbf{e}^{(i+1)}\|_M \leq \|\mathbf{J}^{-1}\|_M \|\mathbf{J}(\mathbf{s}^{(i)}) - \mathbf{B}^{(i)}\|_M \|\mathbf{e}^{(i)}\|_M \leq \left( \frac{1 - e^{\lambda T}}{1 - e^\lambda} \cdot 2e^\lambda \right) \|\mathbf{e}^{(i)}\|_M$$

Thus, in order to achieve linear convergence of the DEER iterates with rate $\beta \in [0, 1)$,

$$\|\mathbf{e}^{(i+1)}\|_M \leq \beta \|\mathbf{e}^{(i)}\|_M \implies \|\mathbf{e}^{(i)}\|_M \leq \beta^i \|\mathbf{e}^{(0)}\|_M,$$

we require that

$$2e^\lambda \cdot \frac{1 - e^{\lambda T}}{1 - e^\lambda} \leq \beta < 1.$$

A simple sufficient condition for satisfying this inequality is

$$e^\lambda \leq \frac{\beta}{2 + \beta} < \frac{1}{3},$$

or,

$$\lambda < -\log(3).$$

$\square$

**Number of Steps to reach basin of quadratic convergence**   Let us assume that there exists $\beta \in [0, 1)$ such that

$$e^{\lambda} \leq \frac{\beta}{2 + \beta}$$

then the number of steps to reach the basin of quadratic convergence is upper bounded as

$$k_Q \leq \log\left(\frac{1}{\beta}\right) \cdot \log\left(\frac{e^{\lambda}||\mathbf{e}^{(0)}||L}{\mu}\right)$$

# G   Alternative Descent Techniques & Worst/Average Complexity

DEER uses the Gauss-Newton algorithm, which converges quadratically near the optimum but can be slow outside this basin. This motivates *inexact* GN methods that guarantee a certain loss decrease per step, such as line-search and trust-region techniques. These trade increased computation and possibly more iterations for faster convergence guarantees.

In practice, we found that plain GN reliably converged quickly to the global optimum in contracting systems, so such safeguards were unnecessary. Still, it is useful to analyze DEER's worst-case path to the quadratic basin.

Many inexact GN variants achieve global convergence from any starting point. These include step-size schemes that approximate a continuous flow [89], trust-regions that bound update size (yielding ELK when applied to DEER [3]), and backtracking line search ensuring loss reduction at each step [21].

One can also use a simpler algorithm outside of the basin of quadratic convergence, and then switch to GN when needed. We will consider this latter option, and choose gradient descent as our simpler algorithm. Because the merit function is PL (see section 3.1), the number of steps required for gradient descent to reach the quadratic convergence region scales as:

$$k_Q \sim \frac{1}{\mu} \cdot \log\frac{||\mathbf{r}^{(0)}||}{\mu}, \tag{36}$$

where $||\mathbf{r}^{(0)}||$ is the residual at initialization. For unpredictable systems, $\mu$ may shrink arbitrarily with increasing sequence length $T$, leading to unbounded growth in the number of optimization steps $k_Q$. By contrast, for predictable systems, $\mu$ remains bounded, implying that the number of optimization steps does not increase with sequence length. Since the cost of sequential evaluation always increases with $T$, DEER can, *even in the worst case*, compute the true rollout faster than sequential evaluation for predictable systems—especially for long sequences. Indeed, assuming the system is contracting with rate $e^{\lambda} < 1$, then the number of steps needed the reach the basin of quadratic convergence is $\mathcal{O}\left(\log ||\mathbf{r}^{(0)}||\right)$.

Thus, if the initial error grows polynomial in $T$, i.e., $||\mathbf{r}^{(0)}|| \propto T^p$, then this implies that the number of gradient descent steps needed to reach the basin of quadratic convergence is only $\mathcal{O}(\log T)$, and thus the total computational time is $\mathcal{O}((\log T)^2)$. In practice, for randomly initialized DEER, we observe $p = 1$.

In practice, we observe that DEER converges much faster than the worst-case analysis (36) would suggest. In particular, we observe that DEER converges in roughly $\log\frac{1}{\mu}$, steps, even for unpredictable systems. This behavior can be explained with a simple "two-phase" model, wherein the DEER iterates move towards the basin of quadratic convergence at a rate which is independent of the PL-constant $\mu$ (see Appendix K.1).

# H   Proof of Size of Basin of Quadratic Convergence

This section provides a proof of Theorem 5:

**Theorem** (Theorem 5). *Let $\mu$ denote the PL-constant of the merit function, which Theorem 2 relates to the LLE $\lambda$. Let $L$ denote the Lipschitz constant of the Jacobian of the dynamics function $J(s)$. Then, $\mu/L$ lower bounds the radius of the basin of quadratic convergence of DEER; that is, if*

$$||\mathbf{r}(\mathbf{s}^{(i)})||_2 \leq \frac{\mu}{L},$$

*then* $\mathbf{s}^{(i)}$ *is inside the basin of quadratic convergence. In terms of the LLE $\lambda$, it follows that if*

$$\|\mathbf{r}(\mathbf{s}^{(i)})\|_2 \leq \frac{1}{a^2 L} \cdot \left( \frac{e^\lambda - 1}{e^{\lambda T} - 1} \right)^2,$$

*then* $\mathbf{s}^{(i)}$ *is inside the basin of quadratic convergence.*

Suppose we are at a point $\mathbf{s}^{(i)} \in \mathbb{R}^{TD}$ (i.e. DEER iterate $i$), and we want to get to $\mathbf{s}^{(i+1)}$. The change in the trajectory obtained from eq. (3) is,

$$\Delta \mathbf{s}^{(i)} := -\mathbf{J}(\mathbf{s}^{(i)})^{-1} \mathbf{r}(\mathbf{s}^{(i)})$$

(where the iteration number will hopefully be clear from context). The merit function is $\mathcal{L}(\mathbf{s}) = \frac{1}{2}\|\mathbf{r}(\mathbf{s})\|_2^2$, so if we can get some control over $\|\mathbf{r}(\mathbf{s}^{(i)})\|_2$, we will be well on our way to proving a quadratic rate of convergence.

First, leveraging the form of the Gauss-Newton update, we can simply "add zero" to write

$$\begin{aligned}
\mathbf{r}(\mathbf{s}^{(i+1)}) &= \mathbf{r}(\mathbf{s}^{(i)} + \Delta \mathbf{s}^{(i)}) \\
&= \mathbf{r}(\mathbf{s}^{(i)} + \Delta \mathbf{s}^{(i)}) - \mathbf{r}(\mathbf{s}^{(i)}) - \mathbf{J}(\mathbf{s}^{(i)}) \Delta \mathbf{s}^{(i)}
\end{aligned}$$

Next, we can write the difference $\mathbf{r}(\mathbf{s}^{(i)} + \Delta \mathbf{s}^{(i)}) - \mathbf{r}(\mathbf{s}^{(i)})$ as the integral of the Jacobian, i.e.

$$\mathbf{r}(\mathbf{s}^{(i)} + \Delta \mathbf{s}^{(i)}) - \mathbf{r}(\mathbf{s}^{(i)}) = \int_0^1 \mathbf{J}\left(\mathbf{s}^{(i)} + \tau \Delta \mathbf{s}^{(i)}\right) \Delta \mathbf{s}^{(i)} \, d\tau.$$

Therefore,

$$\mathbf{r}(\mathbf{s}^{(i+1)}) = \int_0^1 \left( \mathbf{J}\left(\mathbf{s}^{(i)} + \tau \Delta \mathbf{s}^{(i)}\right) - \mathbf{J}(\mathbf{s}^{(i)}) \right) \Delta \mathbf{s}^{(i)} \, d\tau$$

Taking $\ell_2$-norms and using the triangle inequality, it follows that

$$\|\mathbf{r}(\mathbf{s}^{(i+1)})\|_2 \leq \int_0^1 \left\| \left( \mathbf{J}\left(\mathbf{s}^{(i)} + \tau \Delta \mathbf{s}^{(i)}\right) - \mathbf{J}(\mathbf{s}^{(i)}) \right) \Delta \mathbf{s}^{(i)} \right\|_2 d\tau.$$

Now, if we assume that $\mathbf{J}$ is $L$-Lipschitz and use the definition of spectral norm, it follows that

$$\left\| \left( \mathbf{J}\left(\mathbf{s}^{(i)} + \tau \Delta \mathbf{s}^{(i)}\right) - \mathbf{J}(\mathbf{s}^{(i)}) \right) \Delta \mathbf{s}^{(i)} \right\|_2 \leq \tau L \|\Delta \mathbf{s}^{(i)}\|_2^2,$$

and so taking the integral we obtain

$$\begin{aligned}
\|\mathbf{r}(\mathbf{s}^{(i+1)})\|_2 &\leq \frac{L}{2} \|\Delta \mathbf{s}^{(i)}\|_2^2 \\
&= \frac{L}{2} \mathbf{r}(\mathbf{s}^{(i)})^\top \mathbf{J}(\mathbf{s}^{(i)})^{-\top} \mathbf{J}(\mathbf{s}^{(i)})^{-1} \mathbf{r}(\mathbf{s}^{(i)}).
\end{aligned}$$

By definition, $\sqrt{\mu}$ is a lower bound on all singular values of $\mathbf{J}(\mathbf{s}(i))$, for all $i$. Therefore, $\|\mathbf{J}(\mathbf{s}^{(i)})^{-1}\|_2 \leq 1/\sqrt{\mu}$ for all $i$, and it follows that

$$\|\mathbf{r}(\mathbf{s}^{(i+1)})\|_2 \leq \frac{L}{2\mu} \|\mathbf{r}(\mathbf{s}^{(i)})\|_2^2, \tag{37}$$

which is the direct analogy of Boyd and Vandenberghe [31, 9.33]. To reiterate, here $L$ is the Lipschitz constant of $\mathbf{J}$, while $\mu := \inf_{i \in \mathbb{N}} \sigma_{\min}^2\left(\mathbf{J}(\mathbf{s}^{(i)})\right)$.

While this is a quadratic convergence result for GN, this result is not useful unless $\mathbf{r}(\mathbf{s}^{(i+1)})\|_2 \leq \|\mathbf{r}(\mathbf{s}^{(i)})\|_2$ (i.e. would backtracking line search accept this update). However, if we have $\|\mathbf{r}(\mathbf{s}^{(i)})\|_2 < \frac{2\mu}{L}$, then every step guarantees a reduction in $\mathbf{r}$ because in this case

$$\|\mathbf{r}(\mathbf{s}^{(i+1)})\|_2 < \|\mathbf{r}(\mathbf{s}^{(i)})\|_2.$$

Therefore, we have $\|\mathbf{r}(\mathbf{s}^{(j)})\|_2 < \frac{2\mu}{L}$ for all $j > i$. Thus, we have related the size of the basin of quadratic convergence of GN on the DEER objective to the properties of $\mathbf{J}$. Note that with linear dynamics, each $J_t$ is constant in $s$, and so each $J_t$ is 0-Lipschitz. Thus, the basin of quadratic convergence becomes infinite. Intuitively, if $J_t$ doesn't change too quickly with $s$, then DEER becomes a more and more potent method.

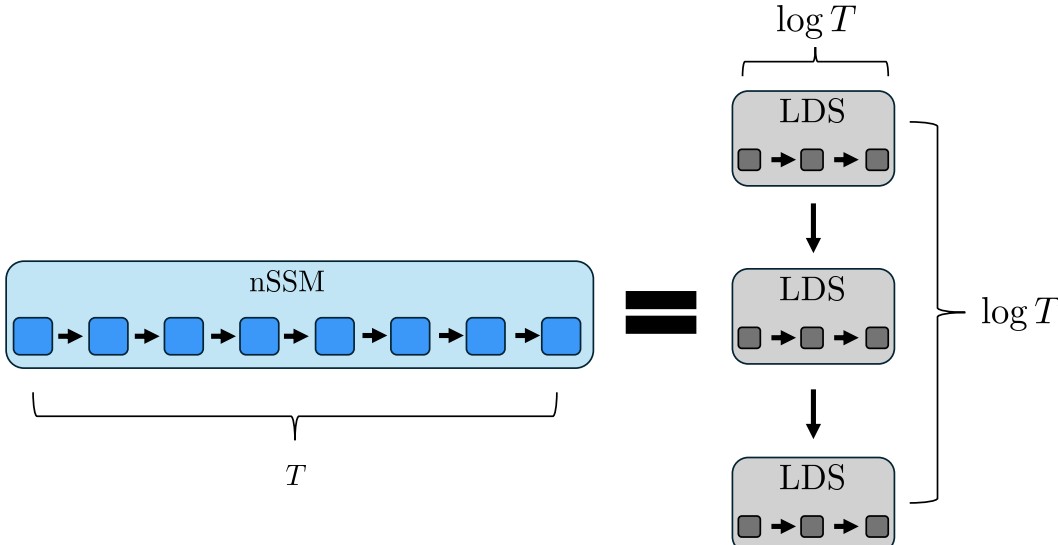

**Figure 4: Equivalence between a contractive nSSM and an $\mathcal{O}(\log T)$ stack of linear state-space models.** Contractivity implies that the nonlinear dynamics can be decomposed into a hierarchy of $\mathcal{O}(\log T)$ layers of linear SSMs, or linear dynamics systems (LDS), each of which can be evaluated in $\mathcal{O}(\log T)$ time by a parallel scan.

## I   Parameterizing nonlinear SSMs to be contractive

In this section, we highlight a practical strategy for speeding up the training of nonlinear state space models (SSMs) based on our theoretical findings.

Our results indicate that nonlinear SSMs with negative largest Lyapunov exponents (LLEs) are efficiently parallelizable. To exploit this during training, one must ensure that the model maintains negative LLEs throughout optimization. One straightforward and effective method to achieve this is by design, through *parameterization*. In particular, by introducing an auxiliary variable to enforce the desired constraint (in this case, negative LLE), and then performing unconstrained optimization on this variable.

This strategy is particularly well-suited to neural network-based SSMs. For example, consider the scalar nonlinear SSM:

$$x_t = \tanh(w x_{t-1} + u_t)$$

To guarantee negative LLE, it suffices to ensure that the Jacobian norm is strictly less than one:

$$|J_t| = |w \cdot \text{sech}^2(w x_{t-1} + u_t)| \leq |w|$$

Thus, enforcing $|w| < 1$ is sufficient. This can be achieved by reparameterizing $w = \tanh(b)$, where $b$ is a trainable, unconstrained auxiliary variable. This guarantees that $w \in (-1, 1)$ for all finite $b$, ensuring contractivity and, hence, negative LLE. A similar argument holds in the multivariate case, using the spectral norm.

## J   Interpreting nonlinear SSMs as stacks of linear dynamical systems

As mentioned in our Discussion in Section 6, an important implication of our results is that a contractive nSSM can be interpreted as a hierarchical composition of linear state-space layers (SSMs), or equivalently, linear dynamical system (LDS) layers. Each layer can be evaluated in $\mathcal{O}(\log T)$ time with a parallel scan, and the total number of layers required scales as $\mathcal{O}(\log T)$. This perspective shows that nonlinear temporal dependencies can be captured through a logarithmic-depth stacking of linear dynamics. Figure 4 provides a schematic illustration of this equivalence.

More explicitly, each iteration of DEER is given by the linear dynamical system

$$s_{t+1}^{(i+1)} = f(s_t^{(i)}) + J_{t+1}(s_t^{(i)}) \left( s_t^{(i+1)} - s_t^{(i)} \right). \tag{38}$$

Therefore, we can interpret each "iteration" $(i)$ of DEER as a sequence-mixing "layer" $(i)$, where the sequence-mixing layer is an input-dependent switching linear dynamical system, like in Mamba [55]. The inputs to "layer" $(i + 1)$ is the state trajectory of the immediately preceding "iteration" or "layer" $(i)$. Because we prove that DEER converges linearly in Theorem 4, it follows that a contractive nSSM can be simulated in $\mathcal{O}(\log T)$ LDS layers of the form shown in eq. (38), assuming the initial error grows polynomially in the sequence length.

## K   Experimental Details and Discussion

All of our experiments use FP64 to, as much as possible, focus on algorithmic factors controlling the rate of convergence of DEER, as opposed to numerical factors. As noted in [3], DEER can be prone to numerical overflow in lower precision. While such numerical overflow can be overcome by resetting `NaNs` to their initialized value, such an approach resets the optimization and leads to rates that are slower than what Gauss-Newton would achieve in infinite precision (exact values in $\mathbb{R}$).

### K.1   Deriving the Empirical Scaling of DEER

In our experiments, we observed that DEER typically converges in $\mathcal{O}(\log(1/\mu))$ steps (see, for example, Figure 2). To understand this scaling behavior, we propose a simple two-phase model of DEER convergence. In the first phase, the iterates approach the basin of quadratic convergence at a linear rate, as guaranteed by Theorem 4. In the second phase, rapid quadratic convergence occurs, typically requiring only one or two steps to reach the true solution (up to floating point precision).

Although Theorem 4 shows that, in unpredictable systems, the overshoot factor may be exponentially large in the sequence length $T$, this reflects a worst-case analysis. In practice, DEER behaves as though the overshoot factor is negligible. To formalize this observation, recall from Theorem 4 that the residuals satisfy the linear convergence bound

$$\|\mathbf{r}_i\| \leq \chi_w \beta^i \|\mathbf{r}_0\|,$$

for some $\beta \in [0, 1)$ and $\chi_w \geq 1$, where $\beta$ is always independent of $T$. In our two-phase model, we assume that $\chi_w$ is also independent of $T$, even when the largest Lyapunov exponent $\lambda$ is positive.

We now upper-bound the number of steps $k$ required to enter the basin of quadratic convergence, whose size scales as $\mu/L$ (as given by (12)). Solving

$$\frac{\mu}{L} = \chi_w \beta^k \|\mathbf{r}_0\| \quad \implies \quad k = \frac{1}{\log \beta} \log\left(\frac{\chi_w L \|\mathbf{r}_0\|}{\mu}\right), \tag{39}$$

we recover the empirically observed logarithmic scaling.

### K.2   Details and Discussion for mean-field RNN experiment

We rolled out trajectories from a mean-field RNN with step size 1 for 20 different random seeds. The dynamics equations follow the form

$$s_{t+1} = W \tanh(s_t) + u_t,$$

for mild sinusoidal inputs $u_t$. We have $s_t \in \mathbb{R}^D$, where in our experiments $D = 100$. Note that because of the placement of the saturating nonlinearity, here $s_t$ represents current, not voltage.

In the design of the weight matrix $W$, we follow Engelken et al. [32]. In particular, we draw each entry $W_{ij} \overset{\text{iid}}{\sim} \mathcal{N}(0, g^2/D)$, where $g$ is a scalar parameter. We then set $W_{ii} = 0$ for all $i$ (no self-coupling of the neurons). A key point of Engelken et al. [32] is that by scaling the single parameter $g$, the resulting RNN goes from predictable to chaotic behavior. While Engelken et al. [32] computes the full Lyapunov spectrum in the limit $D \to \infty$, for finite $D$ we can compute a very accurate numerical approximation to the LLE (cf. Appendix K.6). In Figure 5, we verify numerically that there is a monotonic relationship between $g$ and the LLE of the resulting system, and that the min-max range for 20 seeds is small. Accordingly, when making Figure 2 (Center), we use the monotonic relationship between $g$ and the LLE from Figure 5 to map the average number of DEER steps (over 20 different seeds) needed for convergence for different values of $g$ to the appropriate value of the

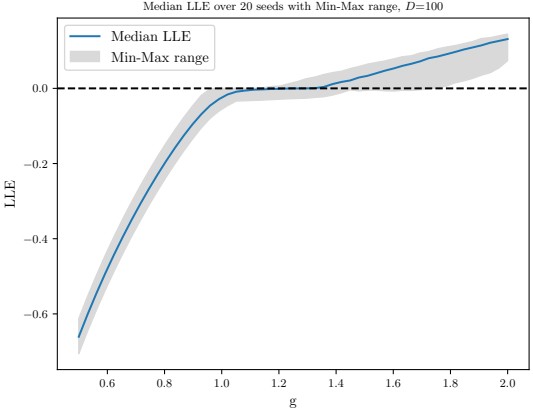

**Figure 5: Robust relationship in mean field RNN between variance parameter $g$ and LLE of the system.** For 20 seeds, we observe a robust and non-decreasing relationship between the scalar parameter $g$ and the LLE of the resulting mean-field RNN. The plot above is made for 50 different values of $g$ from 0.5 to 2.0 (linearly spaced). We estimate the LLE over a sequence length of $T = 9999$.

LLE. We use 50 values of $T$ from 9 to 9999 (log spaced) to make Figure 2 (Center). We highlight $T = 1000$ in Figure 2 (Right).

For the purposes of Figure 2, we define

$$\tilde{\mu} := \left( \frac{e^\lambda - 1}{e^{\lambda T} - 1} \right)^2,$$

i.e. the lower bound on $\mu$ from Theorem 2, with $a = 1$.

In Figure 5, we observe that around $g = 1.2$, the RNNs have LLE around 0, which is the threshold between predictability and chaos. Working with chaotic dynamics in finite precision for long time series led to some interesting difficulties.

First, as discussed in Gonzalez et al. [3], DEER can experience numerical overflow when deployed on unstable systems. While we reset to the initialization (in this experiment we initialized $\mathbf{s}_{1:T}$ with iid draws from $\mathcal{U}[0, 1]$), doing so slows convergence. Thus, many of our runs for $\lambda > 0$ and large $T$ take the maximum number of DEER iterations we allow (we do not allow more than $T$ iterations, as this is the theoretical upper bound for number of DEER iterations before convergence, cf. Proposition 1 of [3]), which helps to explain the slight increase in red space for experiment (center plot of Figure 2) vs. theory (left plot of Figure 2). Note, however, that for $T = 1000$ (the sequence length shown in the right plot), there is no numerical overflow for the DEER trajectories for any of the 20 random seeds or 50 values of $g$ tried.

Second, we observe that for many values of $\lambda$ in the chaotic range, even after the maximum number of DEER steps ($T$) was taken, there was still a large discrepancy between the true sequential rollout and the converged DEER iteration, even though the converged DEER iteration had numerically zero merit function. For example, in Figure 2 (Right), there are a series of points in the top right of the graph that all sit on the line $T = 1000$, and while they have numerically zero merit function value, the converged DEER trajectories are quite different from the true sequential trajectories. The reason for this behavior precisely stems from the fact that for large values of $g$ (equivalently $\lambda$), these mean-field RNNs are chaotic. Even working in FP64, if slight numerical errors are introduced at any time point in the sequence (say $t = 1$), then over the sequence length we can observe exponential divergence from the true trajectories, as illustrated in Figure 6. This experimental observation is complemented by our discussion of why unpredictable systems have excessively flat merit functions in Section 3.2, and provides a numerical perspective on why ill-conditioned landscapes are hard to optimize: if the landscape is extremely flat, many potential trajectories $\mathbf{s}_{1:T}$ can have numerically zero merit function, even in extremely high precision.

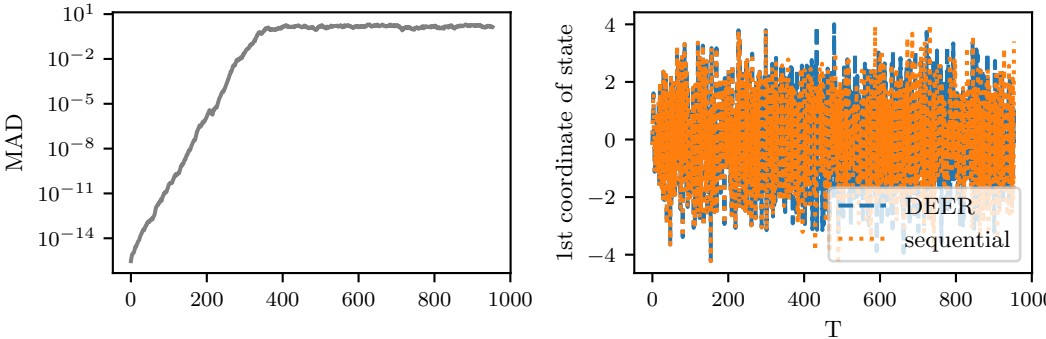

**Figure 6: Chaotic behavior means numerically zero merit function can still be far from sequential trajectory.** For $g = 1.85$ and $T = 1000$, we show the final DEER vs sequential trajectory. The DEER trajectory has merit function (2) numerically equal to zero. However: **(Left)** the mean absolute deviation (MAD) at each time point $t$ between the final DEER iteration $\mathbf{s}_t^{(T)}$ and the sequential rollout $\mathbf{s_t^*}$ grows exponentially. This exponential growth of error is a signature of chaos: compare, for example, with Figure 9.3.5 of Strogatz [10]. The saturation of the error eventually occurs because of the saturating nonlinearity present in the RNN. **(Right)** We visualize the first coordinate of both the final DEER iteration and the sequential trajectory, showing that while they initially coincide, they diverge around $t = 300$.

## K.3 Additional experiment for the mean-field RNN: other optimizers and wallclock time

In this section, we provide further experiments in the setting of the mean-field RNN (Figure 2). In particular, we showcase the generality of our theory beyond DEER (Gauss-Newton optimization), and the practicality of our theory by reporting wallclock times. We consider the setting in the right most panel of Figure 2, where we evaluate a mean field RNN over a sequence length of length $T = 1000$.

**Quasi-Newton and Gradient Descent** Instead of only using Gauss-Newton optimization (DEER) to parallelize the sequence length, we also consider other optimization algorithms (quasi-Newton and gradient descent) to showcase the generality of our theory.

We include a quasi-Newton algorithm proposed in Gonzalez et al. [3] called quasi-DEER. Quasi-DEER simply replaces the $J_t$ defined in eq. (4) with $\text{diag}(J_t)$, and so is also parallelizable over the sequence length with a parallel scan. Furthermore, we also include gradient descent on the merit function, which is embarrassingly parallel over the sequence length. In the top panel of Figure 7, we observe that the number of steps for gradient descent and quasi-DEER to converge also scales monotonically with the LLE, as we expect from Theorem 2. DEER (Gauss-Newton) converges in a small number of steps all the way up to the threshold between predictability and unpredictability ($\lambda = 0$). Intuitively, the performance of the other optimizers degrades more quickly as unpredictability increases because quasi-Newton and gradient descent use less information about the curvature of the loss landscape.

Even though gradient descent was slower to converge in this setting, we only tried gradient descent with a fixed step size. An advantage of a first-order method like gradient descent over a second-order method like Gauss-Newton (DEER) is that the first-order method is embarrassingly parallel (and so with sufficient parallel processors, the update runs in constant time), while DEER and quasi-DEER use parallel scans (and so the update runs in $O(\log T)$ time). Exploring accelerated first-order methods like Adam [90], or particularly Shampoo [91] or SOAP [92] (which are often preferred in recurrent settings like eq. (1))—or in general trying to remove the parallel scan—are therefore very interesting directions for future work.

Sequential evaluation of eq. (1) can also be thought of as block coordinate descent on the merit function $\mathcal{L}(\mathbf{s})$, where the block $s_t \in \mathbb{R}^D$ is optimized at optimization step $(t)$. The optimization of each block is a convex problem: simply minimize $\|s_t - f(s_{t-1}^*)\|_2^2$, or equivalently set $s_t = f(s_{t-1}^*)$. As sequential evaluation will always take $T$ steps to converge, we do not include it in the top panel of Figure 7.

**Wallclock time**  In the bottom panel of Figure 7, we also report the wallclock times for these algorithms to run (our experiments are run on an H100 with 80 GB onboard memory). We observe that the run time of sequential evaluation (green) is effectively constant with respect to $\lambda$. We observe that in the predictable setting, DEER is an order of magnitude faster than sequential evaluation, while in the unpredictable regime, DEER is 1-2 orders of magnitude slower than sequential evaluation. This importance of using parallel evaluation only in predictable settings is a core practical takeaway from our theoretical contributions.

**Further details**  We run the experiment in Figure 7 on a smaller scale than the experiment in Figure 2 (Right). In Figure 7, we consider 5 random seeds for 16 values of $g$ equispaced between $0.5$ and $2.0$. Each wallclock time reported is the average of 5 runs for the same seed. We use a batch size of 1. While DEER (Gauss-Newton) and quasi-DEER effectively do not have a step size (they use a step size of 1 always). For each value of $g$, we ran gradient descent with the following set of step sizes $\alpha$: $0.01, 0.1, 0.25, 0.5, 0.6, 0.7, 0.8, 0.9$, and $1.0$. For each value of $g$, we then pick the step size $\alpha$ that results in the fastest convergence of gradient descent. For the smallest value of $g = 0.5$, we use $\alpha = 0.6$; for $g = 0.6$, we use $\alpha = 0.5$; and for all other values of $g$, we use $\alpha = 0.25$. Future work may investigate more adaptive ways to tune the step size $\alpha$, or to use a learning rate schedule.

We use a larger tolerance of $\mathcal{L}(\mathbf{s})/T \leq 10^{-4}$ to declare convergence than in the rest of the paper (where we use a tolerance of $10^{-10}$) because gradient descent often did not converge to the same degree of numerical precision as sequential, quasi-DEER, or DEER. However, this is a per time-step average error on the order of $10^{-4}$, in a system where $D = 100$ and each state has current on the order of 1. Nonetheless, it is an interesting direction for future work to investigate how to get gradient descent to converge to greater degrees of numerical precision in these settings; and, in general, how to improve the performance of all of these parallel sequence evaluators in lower numerical precision.

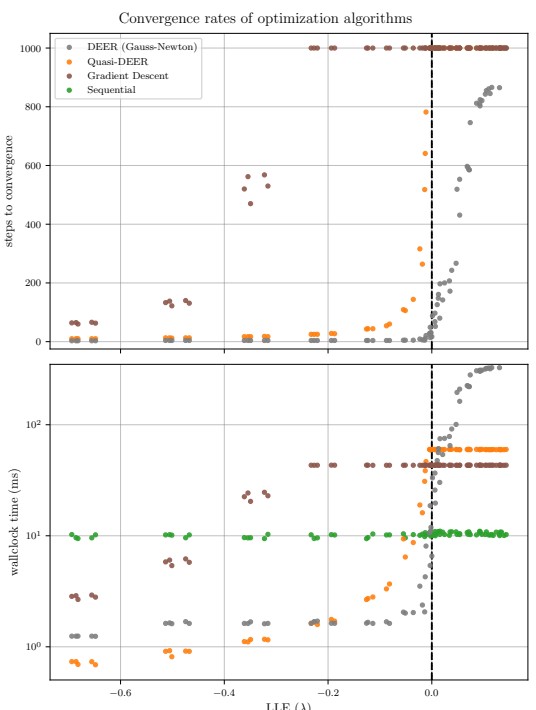

**Figure 7: Convergence rates and wallclock time for many optimizers.** We supplement the mean-field RNN experiment by also considering quasi-Newton and gradient descent methods **(top)**, and recording wallclock time, including for sequential evaluation **(bottom)**.

### K.4 Additional details for the two-well potential

We form the two-well potential for our experiment in Section 5 as a sum of two quadratic potentials. Concretely, we define the potential $\phi$ as the negative log probability of the mixture of two Gaussians, where one is centered at $(0, -1.4)$ and the other is centered at $(0, 1.6)$, and they both have diagonal covariance. In Langevin dynamics [93, 94] for a potential $\phi$, the state $s_t$ evolves according to

$$s_{t+1} = s_t - \epsilon \nabla \phi(s_t) + \sqrt{2\epsilon} w_t,$$

where $\epsilon$ is the step size and $w_t \overset{\text{iid}}{\sim} \mathcal{N}(0, I_D)$. In our experiments, we use $\epsilon = 0.01$. [5] Accordingly, the Jacobians of the dynamics (those used in DEER) take the form

$$J_t = I_D - \epsilon \nabla^2 \phi(s_t).$$

---

[5] Notice that this is a discretization (with time step $\epsilon$) of the Langevin Diffusion SDE $ds(t) = -\nabla \phi(s(t))dt + \sqrt{2}dw(t)$, where $w(t)$ is Brownian motion [95–97].

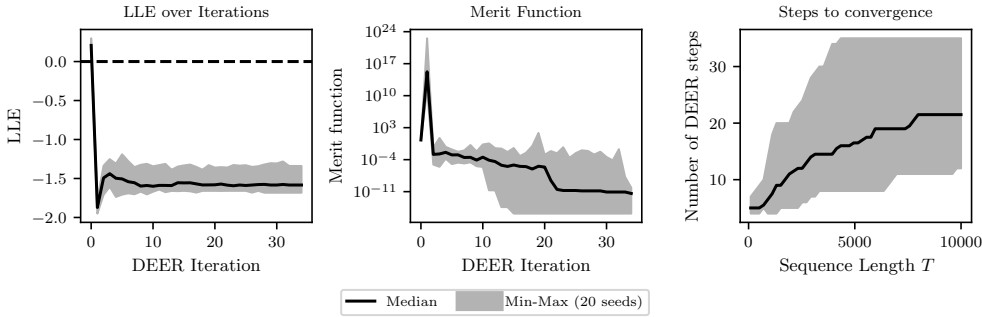

**Figure 8:** In this plot, we provide additional information about the behavior of DEER when rolling out Langevin dynamics on a two-well potential. **(Left)** We observe that across 20 random seeds (including different Langevin dynamics trajectories), the LLE for intermediate DEER iterations becomes negative after the first iteration. Consequently, we observe that the merit function **(Center)** experiences a spike on the very first DEER iteration (following initialization, which was the only trajectory with positive LLE), before trending towards convergence. As the system spends most of its time in contracting regions, we observe **(Right)** that the number of DEER iterations needed for convergence scales sublinearly with the sequence length $T$. We plot the min-max range for 20 seeds, and observe that even out of 20 seeds, the maximum number of DEER iterations needed to converge on a sequence length of $T = 10,000$ is around 35.

As a result, the dynamics are contracting in regions where $\phi$ has positive curvature (inside of the wells, where the dynamics are robustly oriented towards one of the two basins) and unstable in regions where $\phi$ has negative curvature (in the region between the two wells, where the stochastic inputs can strongly influence which basin the trajectory heads towards). We observe that even though there are regions in state space where the dynamics are not contracting, the resulting trajectories have negative LLE. Accordingly, in Figure 3 (Right), we observe that the number of DEER iterations needed for convergence scales sublinearly, as the LLE of all the intermediate DEER trajectories after initialization are negative. These results demonstrate that if the DEER optimization path remains in contractive regions on average, we can still attain fast convergence rates as the sequence length grows.

Moreover, a further added benefit of our theory is demonstrated by our choice of initialization of DEER. Both [1] and [3] exclusively initialized all entries of $\mathbf{s}^{(0)}$ to zero. However, such an initialization can be extremely pathological if the region of state space containing $\mathbf{0}$ is unstable, as is the case for the particular two well potential we consider. For this reason, we initialize $\mathbf{s}^{(0)}$ at random (as iid standard normals).

An important consequence of this experiment is that it shows that there are systems that are not globally contracting that nonetheless enjoy fast rates of convergence with DEER. This fact is important because a globally contractive neural network may not be so interesting/useful for classification, while a locally contracting network could be.

Futhermore, in this experiment we show empirically that Langevin dynamics can have negative LLE (cf. Figure 3). This results suggest that the Metropolis-adjusted Langevin algorithm (MALA), a workhorse of MCMC, may also be predictable in settings of interest, including multimodal distributions.

### K.5   Building Stable Observers for Chaotic Systems

To further demonstrate the applicability of our results—and to validate them in the context of non-autonomous systems—we construct nonlinear observers. Observers are commonly used in science and engineering to reconstruct the full state of a system from partial measurements [33, 34]. As a benchmark, we consider nine chaotic flows from the `dysts` dataset [98]. According to Theorem (2), these systems exhibit poorly conditioned merit function landscapes and are thus not well-suited for parallelization via DEER. If the corresponding observers are stable, then they should be suitable for DEER.

We design observers for these systems using two standard approaches: (1) by directly substituting the observation into the observer dynamics, following Pecora and Carroll [99], or (2) by incorporating the observation as feedback through a gain matrix, as in Zemouche and Boutayeb [100]. We then apply DEER to compute the trajectories of both the original chaotic systems and their corresponding stable observers. As anticipated by Theorem (2), the chaotic systems exhibit slow convergence—often requiring the full sequence length—whereas the stable observers converge rapidly (Figure 9).

As with the two-well experiment, we initialize our guess for $s_t^{(0)}$ as iid standard normals.

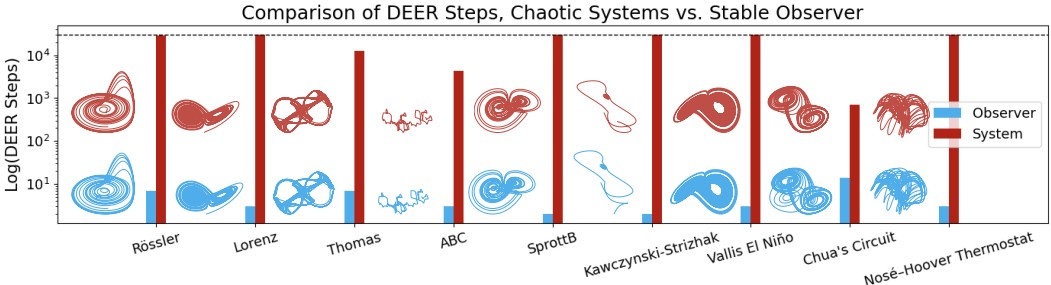

**Figure 9:** Comparison of DEER convergence behavior for original chaotic systems (red) and corresponding stable observers (blue) across nine flows taken from the `dysts` dataset. As predicted by Theorem (2), the chaotic systems converge slowly–often taking the whole sequence length $T$, denoted by the horizontal dashed line–due to poorly conditioned merit landscapes, while the stable observers achieve rapid convergence

.

### K.6  Numerical computation of the discrete-time LLE

The Largest Lyapunov Exponent (LLE), which we often denote by $\lambda$, is defined in Definition 1. However, for long sequences $T$, naively computing it would be numerically unstable. Thus, we use Algorithm 1 to compute the LLE in a numerically stable way. Note that the algorithm nominally depends on the initial unit vector $u_0$. For this reason, we choose 3 different unit vectors (initialized at random on the unit sphere) and average over the 3 stochastic estimates. However, in practice we observe that the estimate is very stable with respect to choice $u_0$, and agrees with systems for which the true LLE is known, such as the Henon and logistics maps.

---

**Algorithm 1** Numerically Stable Computation of Largest Lyapunov Exponent (LLE)

---

1: **Input:** Initial unit vector $u_0$, total iterations $T$
2: **Initialize:** LLE $\leftarrow 0$
3: **for** $t = 1$ to $T$ **do**
4:     Compute evolved vector: $u_t \leftarrow J_t u_{t-1}$
5:     Compute stretch factor: $\lambda_t \leftarrow \|u_t\|$
6:     Normalize vector: $u_t \leftarrow u_t/\lambda_t$
7:     Accumulate logarithmic stretch: LLE $\leftarrow$ LLE $+ \log \lambda_t$
8: **Output:** Estimated LLE $\lambda \leftarrow$ LLE$/T$

---

## L  Discrete and Continuous Time LLE

We provide the definition of the LLE of a discrete-time dynamical system (often called a *map*) in Definition 1. As our paper studies discrete-time SSMs as in (1), this discrete-time definition of LLE makes sense for our setting. However, as many of our experiments involve the discretization of continuous time systems, we want to review how the LLEs of discrete and continuous time systems relate to each other. Helpful references on this topic include [7, 101].

The LLE quantifies what happens to a perturbation[6] $\delta x$ over time. Does its magnitude $\|\delta x\|$ grow or shrink over time? The LLE $\lambda$ is that value that makes eq. (6) hold, i.e.

$$\|\delta x(t)\| \sim e^{\lambda t} \|\delta x(0)\|.$$

This notion of the change in the size of a perturbation $\delta x(t)$ over time—as quantified by the LLE—makes sense for both a discrete-time SSM $x_t = f_t(x_{t-1})$ as well as a continuous time system $\dot{x} = F(x, t)$.

Let us consider how a perturbation $\delta x$ evolves in both a discrete time system $x_t = f_t(x)$ and a continuous time system $\dot{x} = F(x, t)$. An infinitesimal perturbation $\delta x$ is intimately related to derivatives with respect to $x$. Therefore, their *variational equations* are:

$$\text{discrete-time:} \quad \delta x_t = \frac{\partial f_t(x_{t-1})}{\partial x} \delta x_{t-1}$$

$$\text{continuous-time:} \quad \dot{\delta x}(t) = \frac{\partial F}{\partial x}(x(t), t) \, \delta x(t).$$

**Case study: discretizing a continuous-time system**

All of our experiments are the forward Euler discretizations of continuous-time systems. So, in this section, we work out what happens to the LLE in such a setting. Let's consider running our continuous-time setting for a length of time $T$.

In this setting, if we discretize by timestep $\Delta t$, our resulting discrete-time map $f$ is given by

$$f_{t+\Delta t}(x_t) = x_t + \Delta t F(x_t, t).$$

Therefore, it follows that

$$J_{t+\Delta t} := \frac{\partial f}{\partial x}(x_t) = I_D + \Delta t \frac{\partial F}{\partial x}(x_t).$$

We can naturally define the continuous-time LLE as the limit of the products of these $J_t$ as $\Delta t \to 0$, i.e.

$$\lambda_c := \lim_{\Delta t \to 0} \frac{1}{T} \log \left\| \prod_{k=1}^{N} J_{k\Delta t} \right\|,$$

where the number of discrete time-steps $N$ is given by $N := T/\Delta t$.

Notice that this is extremely similar to the definition of the discrete-time LLE $\lambda_d$ we gave in Definition 1, except division is occurring by the length of the time window $T$ instead of the number of discrete steps taken $N = T/\Delta t$. (Of course, $N = T$ if $\Delta t = 1$.)

Therefore, if we have a discretization of a continuous-time system, and naively plug into our discrete-time LLE defined in Definition 1 (i.e., divide by number of discrete-time steps $N$ instead of the length of the time window $T$), the resulting $\lambda_d(\Delta t)$ will satisfy

$$\lambda_c = \frac{\lambda_d(\Delta t)}{\Delta t}.$$

Note that naively plugging in this discrete-time estimator would therefore result in a different estimate of the LLE depending on the size of the time-step $\Delta t$. However, in most of this paper we still report the naive discrete-time LLE $\lambda_d(\Delta t)$ as this is the quantity relates to $\mu$ via $\mathbf{J}$. The exception to our convention is in Table 1, where we report our estimates $\lambda_c$ to better coincide with the intuitions and expectations of readers with backgrounds in continuous time systems. For all systems in Table 1, we use a step size of 0.01, and so one can translate from the reported continuous time LLEs to discrete time LLEs by dividing by 100.

Ultimately, dividing by $\Delta t$, which is positive, does not change the sign of $\lambda$, i.e. whether or not the system is predictable or unpredictable.

---

[6]technically a *virtual displacement*, cf. Lohmiller and Slotine [7].

