# OpenReview forum: "Predictability Enables Parallelization of Nonlinear State Space Models"
_NeurIPS.cc/2025/Conference — NeurIPS 2025 poster_

### Official Review · Reviewer_HUX6 · 2025-06-25

**Clarity:** 4
**Significance:** 3
**Originality:** 3
**Rating:** 5
**Confidence:** 4

**Summary:**

The paper presents a theoretical framework relating the convergence speed of parallel evaluation in nonlinear state space models to the predictability of the system dynamics through the conditioning of a merit function. The authors support the theory with several experiments showing strong agreement between empirical results and theoretical predictions.

**Questions:**

1. What are the axis labels in Figure 1? What do the colors of the contour plot mean? Its unclear what is being shown in each panel.
2. In the experiments, it is only shown the number of steps to converge for parallel evaluation. How do these results compare to the speed of sequential evaluation?
3. What is the cost of computing the LLE? What is the memory footprint and runtime complexity? Is it cheap enough to compute during a training loop when training an RNN?
4. Is the magnitude of the LLE important for interpretation? It seems from Table 1 that the same number of newton steps required can correspond to very different LLE scores.
5. How can this theory be exploited to improve the speed of training nonlinear state space models?

**Ethical Concerns:**

["NO or VERY MINOR ethics concerns only"]

**Final Justification:**

The authors provide additional runtime experiments and a more detailed discussion on the cost and practicality of computing LLE, which should be included in the main paper or supplementary material. I also appreciate the discussion on how the proposed theory can be applied to improve RNN training efficiency. My questions and concerns have been fully addressed, and I have increased my score to accept.

**Limitations:**

yes

**Quality:**

3

**Strengths And Weaknesses:**

Strength
1. Analytic results are theoretically sound and a very nice contribution to the current literature on parallelizing nonlinear SSMs.
2. Empirical experiments are well thought out and directly validate the proposed theory.
3. The use tools from dynamical systems to characterize parallel evaluation of RNNs is a novel contribution.
4. The text is well-written and explanations are intuitive and easy to follow.


Weakness
1. **No comparison with sequential speed.** The experiments report only the number of parallel steps to convergence, without comparing against sequential evaluation. It is unclear whether the theory offers a practical criterion for deciding between sequential and parallel evaluation. Reporting wall-clock times for sequential and parallel approaches for each experiment would help clarify this connection between LLE and the decision to use parallelism.
2. **No benchmarking of LLE computation cost.** While the LLE is proposed as a criterion for when parallel evaluation is feasible, the practicality of computing the LLE itself is unclear. What is the computational complexity of estimating the LLE, and can it be efficiently computed in the context of large-scale models?
3. **Unclear connection to RNN training efficiency.** Although the experiments empirically validate aspects of the theory, it remains unclear how this theory can be used to accelerate training of nonlinear sequence models such as RNNs. I would like to see either a concrete demonstration of improved training speed using the proposed approach, or at least a discussion/outline of how the theory could inform better training strategies for recurrent models.

---

> ### Author Rebuttal · Authors · 2025-07-31
>
> We thank the reviewer for taking the time to review our paper. We are especially pleased by the comments that our contribution is “very nice” and “novel”; our analysis is "theoretically sound”;  our experiments are “well thought out” and “directly validate” our theory; and our text is “well-written”. We also appreciate the thoughtful feedback, which we have addressed and incorporated in our revision.
>
> ## Comparison with sequential speed
>
> We have added wall-clock times for both parallel and sequential evaluation in our experiments. Our experiments use an H100 with 80GB onboard memory, and we average our results over five runs.
>
> For example, in the setting of Experiment 1 (Figure 2, right) in our paper, we have added wallclock time comparisons, as follows.
>
> ### Wall‑clock time (ms) by LLE for Newton & sequential evaluation
> | LLE               |   Newton |   sequential evaluation |
> |:------------------|---------:|------------------------:|
> | -0.7              |     1.22 |                    9.56 |
> | -0.5              |     1.59 |                    9.65 |
> | -0.35             |     1.59 |                    9.17 |
> | -0.2              |     1.61 |                    9.34 |
> | -0.1              |     1.73 |                    9.46 |
> | -0.1 to 0         |     6.84 |                    9.82 |
> | 0 to 0.05         |    53.59 |                    9.92 |
> | 0.05 to 0.1       |   233.33 |                    9.97 |
> | 0.1 to 0.14       |   325.71 |                   10.08 |
> | greater than 0.14 |   345.47 |                   10.09 |
>
> We observe that in the predictable setting, Newton is an order of magnitude faster than sequential evaluation, while in the unpredictable regime, Newton is 1-2 orders of magnitude slower than sequential evaluation. This importance of using parallel evaluation only in predictable settings is a core practical takeaway from our theoretical contributions.
>
> ## Discussion of Largest Lyapunov Exponent
>
> ### LLE computation cost
>
> We use a standard algorithm (Algorithm 1 in Appendix J.5) for computing the largest Lyapunov exponent (LLE). This algorithm propagates a unit vector through a sequence of $T$ Jacobians, normalizing at each step for numerical stability of the estimation (see e.g., Section 3.1 of Politi, A., & Pikovsky, A. (2010). *Lyapunov Exponents: A Tool to Explore Complex Dynamics.*).
>
> Algorithm 1 has a memory footprint of only $O(D^2)$. We can also use Jacobian vector products (JVPs) to reduce the memory footprint to $O(D)$. Using these JVPs, the computational complexity is $O(TD)$.
>
> ### Can LLE be computed and used during training?
>
> The LLE is indeed cheap enough to compute during an RNN training loop, as demonstrated in [1] where _all_ Lyapunov exponents—not just the top one—are computed during training and used to regularize the RNN away from chaotic solutions.
>
> Finally, we emphasize that in many scenarios, computing the LLE during training is unnecessary. For instance, if a system is known or constrained to be contracting/stable *a priori*, our results imply that it is parallelizable regardless of the exact LLE value. This situation arises, for example, in modern state space models and many RNN architectures (see [2-3] below).
>
> [1] Engelken, Rainer. "Gradient flossing: Improving gradient descent through dynamic control of Jacobians." Advances in Neural Information Processing Systems 36 (2023): 10412-10439.
>
> [2] Gu, Albert, and Tri Dao. "Mamba: Linear-time sequence modeling with selective state spaces." arXiv:2312.00752 (2023).
>
> [3] Krotov, D. A new frontier for Hopfield networks. Nat Rev Phys 5, 366–367 (2023).
>
> ### Interpreting the magnitude of the LLE
> We thank the reviewer for raising this interesting and subtle point. In addition to the Lyapunov Exponent (LLE), our theory (specifically equation 9) shows that the “overshoot” parameter $a$ also plays a role in determining the flatness of the merit function. In principle, two systems could have the same LLE value but different overshoot parameters, resulting in different numbers of optimization steps. We focus on LLE rather than overshoot because the dependence on LLE is exponential, whereas the dependence on overshoot is only linear. Therefore, for large sequences, it is primarily the LLE that determines whether parallelization via merit function optimization is feasible. We have revised the text to better highlight this important point.
> ## Application to RNN training efficiency
>
> We thank the reviewer for the insightful question. Below, we highlight a practical strategy for speeding up the training of nonlinear state space models (SSMs) based on our theoretical findings.
>
> Our results indicate that nonlinear SSMs with negative largest Lyapunov exponents (LLEs) are efficiently parallelizable. To exploit this during training, one must ensure that the model maintains negative LLEs throughout optimization. One straightforward and effective method to achieve this is by design, through **parameterization**. In particular, by introducing an auxiliary variable to enforce the desired constraint (in this case, negative LLE), and then performing unconstrained optimization on this variable.
>
> This strategy is particularly well-suited to neural network-based SSMs. For example, consider the scalar nonlinear SSM:
>
> $x_t = \tanh(w x_{t-1} + u_t).$
>
> To guarantee negative LLE, it suffices to ensure that the Jacobian norm is strictly less than one:
>
> $ |J_t| = |w \cdot \text{sech}^2(w x_{t-1} + u_t)| \leq |w|.$
>
> Thus, enforcing $|w| < 1$ is sufficient. This can be achieved by reparameterizing $w = \tanh(b)$, where $b$ is a trainable, unconstrained auxiliary variable. This guarantees that $w \in (-1, 1)$ for all finite $b$, ensuring contractivity and, hence, negative LLE.
>
> This approach has precedent in the literature on training stable RNNs [e.g., Rainer et al., Manchester et al., and Farsang et al.]. We particularly highlight the concurrent work of Farsang et al. (arXiv:2505.21717), which scales up a nonlinear SSM using parallel training with DEER. Farsang et al explicitly parameterizes their SSM to be contractive (see their Appendix A.1). However, they do not make the connection that this contractivity is precisely why DEER enjoys fast convergence throughout training. Ensuring negative LLE via parameterization enables models to remain parallelizable throughout training, providing a pathway to faster and more scalable learning.
>
> We have more heavily emphasized how enforcing stability in the design of nonlinear RNNs allows for their efficient parallelization and scalability. Our work plays an important role from moving from heuristics to firm theoretical grounding for the importance of stability in the design of efficiently parallelizable nonlinear SSMs.
>
>
> ## Clarifying Figure 1
>
> Figure 1 is our banner schematic, so we appreciate feedback on it. While this year’s NeurIPS does not allow us to upload images, we provide answers to the reviewer’s first question and have adjusted the schematic to be more clear (especially adding axis labels).
>
> Figure 1 is a schematic illustrating Theorem 2, which proves that the predictability $\lambda$ of a dynamical system (illustrated by the top row, labelled “state space”) determines the conditioning $\mu$ of its merit function landscape (illustrated by the bottom row, labelled “merit function”). The left column (blue font), shows that predictable/contracting systems result in well-conditioned loss landscapes. The right column (red font) shows that unpredictable systems result in ill-conditioned (flat) landscapes.
>
> The system we used to generate this banner schematic is a vanilla RNN with a hidden state of dimension 1; we can adjust the scalar weight to result in either predictable or unpredictable dynamics.
>
> The top row shows the trajectories over time of these RNNs. Thus, the x-axis is time, while the y-axis is the value in state space. The gray arrows show flow fields based on the particular value in state space, while the solid lines (blue for predictable, red for unpredictable) indicate the trajectories resulting from two different initial conditions. Trajectories from predictable systems converge over time, while trajectories from unpredictable systems diverge over time.
>
> The bottom row (the contour plots) show the resulting merit function landscapes (see equation 2 for the definition of the merit function). While the merit function is a function of $\mathbf{s} \in \mathbb{R}^T$, for visualization we pick two time steps for the axis labels. In this particular visualization, the x-axis is $s_{50}$ and the y-axis is $s_{51}$. The color of the contour plot is the value of the merit function for particular values of $\mathbf{s}$, with darker color indicating a lower value of the merit function (pitch black correspond to the minimum $\mathcal{L}(\mathbf{s}^*) = 0$). The blue/red dotted lines indicate the optimization trajectory of DEER for these two RNNs. In the predictable setting, the merit function landscape is well-conditioned (bowl shaped), and DEER quickly moves to the minima (the blue star). In the unpredictable setting, the merit function landscape is ill-conditioned (has a long, flat, narrow valley), and DEER oscillates around the minima (the red star).
>
> We have added axis labels and expanded the caption to improve the readability of this banner schematic.

---

> > ### Comment · Reviewer_HUX6 · 2025-08-01
> >
> > Thank you for your thorough responses! The authors provide additional runtime experiments and a more detailed discussion on the cost and practicality of computing LLE, which should be included in the main paper or supplementary material. I also appreciate the discussion on how the proposed theory can be applied to improve RNN training efficiency. My questions and concerns have been fully addressed, and I have increased my score to accept.

---

> > > ### Author Response · Authors · 2025-08-01
> > > **Thank you for advocating for acceptance of our paper!**
> > >
> > > Thank you for advocating for acceptance of our paper!

---

### Official Review · Reviewer_XHnm · 2025-07-02

**Clarity:** 4
**Significance:** 4
**Originality:** 3
**Rating:** 5
**Confidence:** 3

**Summary:**

The paper establishes a direct link between how predictable the system is, measured by its largest Lyapunov exponent, and the minimum singular value of the merit function, which is a set of sequential estimation errors. Specifically, the authors show that if small perturbations are diminished over time, which is defined by the largest Lyapunov exponent, the authors also show that the merit function is well-conditioned, which allows for parallel evaluation and utilizing algorithms such as DEER. On the contrary, if perturbations are not diminished over time, the authors also show that it will take an exponential number of steps for convergence. The authors showcase the theoretical results in mean-field RNNs and Langevin dynamics to show how DEER scales with stable and chaotic systems.

**Questions:**

The theoretical conditions developed by the paper are well-defined and well-presented. However, I would like to ask some questions about the algorithm.

This does not affect the theoretical results, but I was curious whether there are existing bounds between the finite-time estimates of the Lyapunov exponent versus the infinite-horizon Lyapunov exponent used in the theoretical results.

I believe the authors require a global Lipschitz condition to hold for global convergence of the results, which is defined in Theorem 4. I wonder if it's possible to establish similar results with local Lipschitz assumptions while understanding that the convergence would depend on the initialization.

If the model is only locally Lipschitz, does this imply that the Lyapunov exponent is positive, and hence, it's computationally expensive to perform parallel estimation? The authors consider systems that satisfy Lipschitz conditions globally, but I was curious if these conditions do not hold with locally Lipschitz systems, which is very common in non-linear systems.

**Ethical Concerns:**

["NO or VERY MINOR ethics concerns only"]

**Final Justification:**

I think the paper can be accepted for publication. The authors addressed my comments and the other reviewers comments well, and I initially recommended the acceptance of the paper.

**Limitations:**

yes

**Quality:**

4

**Strengths And Weaknesses:**

The paper provides a well-defined notion of stability of the non-linear systems and the conditioning of the merit functions, which provides a clear signal on the parallelizability of the inference of non-linear systems, such as using algorithms like DEER.

The only weakness that I can find is, the paper assumes that the non-linear systems satisfy global Lipschitz conditions, which may not hold for non-linear systems that often appear in practice. However, the global Lipschitz conditions may be necessary for parallel inference with the metric functions, and the authors may clarify this in the rebuttal phase.

---

> ### Author Rebuttal · Authors · 2025-07-31
>
> We thank the reviewer for their thoughtful and positive review, and for their interesting questions about Lipschitzness and the Largest Lyapunov Exponent. We discuss the points they raise below.
>
> > The only weakness that I can find is, the paper assumes that the non-linear systems satisfy global Lipschitz conditions, which may not hold for non-linear systems that often appear in practice.
>
> Many of the results in our paper do not depend on Lipschitz conditions (Theorem 1, Theorem 2, and Theorem 4). In particular, we emphasize that our main result, Theorem 2—which establishes a direct link between how predictable the system is, measured by its largest Lyapunov exponent, and the minimum singular value of the merit function Jacobian—does not have Lipschitzness as an assumption.
>
> However, the reviewer is correct that Theorem 5 does require a Lipschitz assumption. We agree that an interesting direction for future work could be to investigate if Lipchitzness conditions for this theorem could be relaxed.
>
> Finally, while we agree that there are important non-linear systems that are not Lipschitz, we note that many systems in deep learning are Lipschitz by design. In particular, if the transition function is some combination of multiplication by a finite dimensional matrix and the application of a Lipschitz nonlinearity (such as a ReLU or saturating nonlinearities), then that transition function is also Lipschitz (cf. eg. Szegedy et al, “Intriguing properties of neural networks”).
>
> > This does not affect the theoretical results, but I was curious whether there are existing bounds between the finite-time estimates of the Lyapunov exponent versus the infinite-horizon Lyapunov exponent used in the theoretical results.
>
> We thank the reviewer for raising this interesting question.
>
> This topic has been addressed in the literature. For example, see Sections 4 ("Error Analysis") and 5 of [1] below, where bounds are derived between the finite-time Lyapunov exponents and their infinite-time limits. These results apply to all Lyapunov exponents, not just the largest one.
>
> An alternative approach to this question is to consider it probabilistically. In particular, where one can assert—with high probability—that the finite-time estimate deviates from the infinite-time exponent by no more than a specified amount. These bounds are usually found in the context of ergodic or random dynamical systems and are framed using large deviation principles. See reference [2] below and references therein for examples of this second approach.
>
> [1] Dieci, L., Russell, R. D., & Van Vleck, E. S. (1997). On the computation of Lyapunov exponents for continuous dynamical systems. SIAM journal on numerical analysis, 34(1), 402-423.
>
> [2] Prasad, A., & Ramaswamy, R. (1999). Characteristic distributions of finite-time Lyapunov exponents. Physical Review E, 60(3), 2761.
>
> > However, the global Lipschitz conditions may be necessary for parallel inference with the metric functions, and the authors may clarify this in the rebuttal phase.
> and
> > I believe the authors require a global Lipschitz condition to hold for global convergence of the results, which is defined in Theorem 4. I wonder if it's possible to establish similar results with local Lipschitz assumptions while understanding that the convergence would depend on the initialization.
>
> We thank the reviewer for the opportunity to clarify this point. Importantly, the proof of Theorem 4 does not require a global Lipschitz condition. However, a global Lipschitz condition is used elsewhere in the paper (e.g., in Theorem 5). We have revised the manuscript to clarify this distinction.
> We agree that relaxing the global Lipschitz requirement in favor of a local one, with convergence guarantees that depend explicitly on the initialization, is an important (and promising!) direction. While we have not pursued this in the current work, we are optimistic that our results could be extended in this way.
> > If the model is only locally Lipschitz, does this imply that the Lyapunov exponent is positive, and hence, it's computationally expensive to perform parallel estimation? The authors consider systems that satisfy Lipschitz conditions globally, but I was curious if these conditions do not hold with locally Lipschitz systems, which is very common in non-linear systems.
> We thank the reviewer for this interesting question. The Lipschitzness of a model and its Lyapunov exponents can in fact be decoupled, in the sense that a model only being locally Lipschitz does not imply a positive Lyapunov exponent, and vice versa.
>
> This can be seen with an example. Consider the system:
>
> $x_t = (1-h) x_{t-1} - h x^3_{t-1}$.
>
> Where $0 < h \ll 1$. Note that this system is the Euler discretization of the stable, continuous-time dynamics $dx/dt = -x - x^3$, with Euler step-size h.
>
> This system is not globally Lipschitz, because its Jacobian is
> $J(x_{t-1}) = (1- h) - 2 h x^2_{t-1}$
>
> Since $J’(x) = -4hx$ can become unbounded with increasing $x$, the Jacobian is not Lipschitz.
>
> Let us now evaluate the LLE around the origin. When $x = 0$, we have that
>
> $|J(0)| = |1- h| < 1 $.
>
> Thus, the origin is a locally stable trajectory and thus has negative LLE. Indeed, for this system, one can show that if the initial condition $x_0$ is drawn from a finite ball of radius $R$ around the origin, then there exists a sufficiently small $h(R)$ such that: 1) the ball is forward-invariant—that is, all trajectories remain within it for all time; and 2) the Jacobian along all trajectories has absolute value less than one, implying that the system is contracting and the largest Lyapunov exponent (LLE) is negative for all trajectories.
> Thus, our results (in particular equation 9 and Theorem 4) hold for systems which are not globally Lipschitz.

---

> > ### Comment · Reviewer_XHnm · 2025-08-04
> >
> > Thanks a lot for the detailed response and the clarification examples of the underlying assumptions about Lipschitz conditions and the stability of the trajectories. I am still happy with my recommendation of accepting this paper.

---

> > > ### Author Response · Authors · 2025-08-05
> > > **Thank you for advocating to accept the paper!**
> > >
> > > We are happy to have answered your questions! Thank you for advocating to accept the paper!

---

### Official Review · Reviewer_XTsi · 2025-07-03

**Clarity:** 4
**Significance:** 3
**Originality:** 3
**Rating:** 5
**Confidence:** 3

**Summary:**

The paper investigates the condition in which nonlinear state space models can be efficiently
parallelized. Although recent methods like DEER (non-linear Differential Equation as fixed point
itERation) where the model was able to apply parallel processes to efficiently train sequential
models, what factors determine the convergence speed of this optimization remains unclear.
Therefore, the authors demonstrate that a predictable system, measured by the Largest
Lyapunov Exponent, corresponds to well-conditional optimization problems that can be solved in
parallel process, wherares unpredictable systems lead to ill-conditioned problems that converge
too slowly.

**Questions:**

NA

**Ethical Concerns:**

["NO or VERY MINOR ethics concerns only"]

**Limitations:**

Yes

**Quality:**

3

**Strengths And Weaknesses:**

The paper’s strength lies in its main contribution of connecting the Largest Lyapunov Exponent
to PL conditioning in optimization problems. The formalization of predictable and unpredictable
systems was also creative and paved the way for the author’s main work. The proof was also
clear, and the author used a strong intuition, such as the "flatness" of the merit landscape for
chaotic systems, to explain complex mathematical results. The work has practical significance
for state-space models and recurrent neural networks to operate on parallel hardware.

However, the paper’s theoretical convergence results and analysis is closely tight with the
DEER algorithm. The authors did not fully explore other methods.
Additionally, the work has a niche relevance since a lot of the deep learning community has been shifted to inherently
parallel architectures like Transformers.

---

> ### Author Rebuttal · Authors · 2025-07-31
>
> Firstly, we thank the reviewer for taking the time to review our submission and for their positive comments! We were particularly pleased by the comments on the creativity, clarity, and practicality of our work.
>
> The reviewer provided two main areas for improvement: an exploration of other optimization algorithms, and a better discussion of the relevance of our theory in the context of transformers’ popularity. By incorporating this feedback, we believe that we have strengthened our submission. We discuss in detail below.
>
> ## Our theory also applies to other optimizers like gradient descent and quasi-Newton methods
>
> We agree that it is important to emphasize the broad applicability of our theoretical contribution to optimization techniques beyond the Gauss-Newton method (DEER).
>
> To recapitulate, Theorem 2 establishes a relationship between the conditioning of the merit function and the Largest Lyapunov Exponent (LLE) $\lambda$. Specifically, we show that the Polyak–Łojasiewicz (PL) constant $\mu$ of the merit function increases monotonically with $\lambda$. The PL constant $\mu$ quantifies the conditioning of the optimization problem—i.e., how "well-conditioned" the merit function landscape is—while the LLE $\lambda$ measures the predictability of the underlying dynamical system.
> Theorem 2 thus demonstrates that systems with low predictability (high $\lambda$) exhibit poorly conditioned merit function landscapes, whereas more predictable systems (low $\lambda$) yield well-conditioned landscapes. Importantly, this result concerns *the landscape of the merit function itself*, independent of any specific optimization algorithm. Therefore, our theory is not limited to the Gauss-Newton method (DEER), but extends to a broad class of optimization methods and settings.
> For instance, as discussed on line 131, the PL condition guarantees that gradient descent achieves a linear convergence rate on the merit function landscape. However, the rate constant depends on the value of $\mu$, which we observe to become prohibitively small in the case of unpredictable systems—leading to extremely slow overall convergence. This highlights that our theoretical results are not limited to Gauss-Newton method (DEER) but also extend to gradient descent. We have clarified and expanded on this point in the revised manuscript to better emphasize the broad applicability of our theoretical contributions across optimization algorithms.
> We have also strengthened our experimental evaluation by including results for both **gradient descent** and a **quasi-Newton method**. Specifically, we revisited our first experiment involving the mean-field RNN (see Figure 2 in the paper). In the setting depicted in the rightmost plot of this figure (state dimension $D = 100$, sequence length $T = 954$), we conducted an additional experiment comparing the convergence rates of Newton’s method, a quasi-Newton method, and gradient descent. The results are summarized in the following table (unfortunately, images are not permitted in this year’s NeurIPS rebuttals):
>
> ###  Iterations until convergence
> | LLE               |   Newton |   quasi-Newton |   gradient descent |
> |:------------------|---------:|---------------:|-------------------:|
> | -0.7              |        3 |             10 |                 31 |
> | -0.5              |        4 |             13 |                 58 |
> | -0.35             |        4 |             17 |                206 |
> | -0.2              |        4 |             26 |                435 |
> | -0.1              |        4 |             66 |                954 |
> | -0.1 to 0         |       18 |            674 |                954 |
> | 0 to 0.05         |      147 |            954 |                954 |
> | 0.05 to 0.1       |      642 |            954 |                954 |
> | 0.1 to 0.14       |      895 |            954 |                954 |
> | greater than 0.14 |      954 |            954 |                954 |
>
> For all three optimizers, we observe that the number of steps to convergence increases monotonically with the value of the LLE. This occurs because, as we proved in Theorem 2, the conditioning $\mu$ of the merit function landscape degrades monotonically as the LLE $\lambda$ increases.
>
> ## Parallelizing sequential operations is broadly important
>
> We thank the reviewer for emphasizing the need to highlight the importance and relevance of our work in the context of transformers. While the original DEER paper (Lim et al) focused exclusively on RNNs and Neural ODEs for sequence modeling, we should have emphasized that DEER can apply to any Markovian system. In our revisions, we have more heavily emphasized the importance of parallelizing sequential computation generally, and not just in sequence modeling. The below text indicates our reframing of the contextualization:
>
> Markovian systems involving sequential computation abound across deep learning, probabilistic modeling, and scientific computing. In particular:
>
> ### 1. Parallelization over *depth* is relevant for transformers
>
> We agree that transformers are an extremely important architecture, and that the attention mechanism is parallelized *over the sequence length*. However, almost all transformers consist of multiple transformer blocks that must be applied sequentially *in depth*. Therefore, sequential operations also occur in transformers. As transformers become increasingly deep, parallelizing Transformers over depth will become even more important. In fact, preliminary efforts to this effect have already been attempted (Calvo Gonzalez et al, “Leveraging the true depths of LLMs”, ‘25). Another important trend is “recurrent depth” in transformers. “Recurrent depth” treats test-time-compute/latent space “thinking” as an RNN where the time step function is a transformer block (see Schone et al, “Implicit Language Models are LLMs”, ICML ‘25 Spotlight;  and Geipeng et al, “Scaling up Test-Time Compute with Latent Reasoning: A Recurrent Depth Approach”, ‘25). We believe DEER could play a key role in enabling efficient, complete parallelization for such models, especially those that involve iterative or recurrent computation over depth.
>
> ### 2. Stateful models are relevant across ML
>
> A huge range of important models in deep learning and machine learning involve stateful computation. These include the sampling pass of a diffusion model, the forward pass of a feedforward network, and the state of a reinforcement learning agent. Moreover, stateful operations are important in probabilistic modelling, such as Markov Chain Monte Carlo (MCMC). In fact, our results in Figure 3 about the parallelizability of Langevin dynamics is highly suggestive about applications to MCMC because MALA (Metropolis-adjusted Langevin algorithm) is an important MCMC workhorse. We have better emphasized this connection in our revision.
>
> ### 3. Stateful models are relevant in scientific computing
>
> Finally, the ability to parallelize seemingly sequential computation is important in scientific computing broadly. Notable examples include the solving of differential equations and computing the trajectories of systems under various forms of control. In fact, our results in Table 1 about the difficulties of parallelizing chaotic observers, and the ability to parallelize stable observers, relate to both these fields (solving ODEs and controlling systems). We have more heavily emphasized these connections in our revision.
>
> ### In summary
>
> We have expanded our introduction to appropriately motivate the importance of sequential operations as a primitive across machine learning and scientific computing. We situate our contribution as providing an important theoretical basis for determining in which settings sequential computation can be efficiently parallelized.

---

### Official Review · Reviewer_HYQS · 2025-07-04

**Clarity:** 3
**Significance:** 3
**Originality:** 2
**Rating:** 4
**Confidence:** 3

**Summary:**

The paper presents a theoretical framework demonstrating that a system’s predictability governs the number of optimization steps needed for evaluation. This relationship is important for assessing the efficiency of parallel computing. The authors validate their approach by analyzing the convergence behavior of the Gauss-Newton DEER algorithm using a worst-case bound on optimization steps.

**Questions:**

- How does the degree of predictability (i.e., sensitivity to perturbations) affect the results?

- How does the system’s dimension impact the results?

- Could you clarify the example in lines 177–182? You state residuals for later times are assumed to be zero, but later mention residuals grow over time. What exactly is assumed in that example?

**Ethical Concerns:**

["NO or VERY MINOR ethics concerns only"]

**Final Justification:**

I thank the authors for their response and believe that the clarifications, additional comparisons, and discussion points will strengthen the paper. I increased my score to 4.

**Limitations:**

yes

**Quality:**

3

**Strengths And Weaknesses:**

## Strengths
- Clear presentation of the problem, motivation, and overall approach (though the organization could be improved, see below).
- The motivation behind the theory is well explained.
- The authors provided proofs where appropriate.
- Demonstrated the model using the DEER algorithm.

## Weaknesses
- The evaluation is limited to the DEER algorithm. Including comparisons to additional algorithms will help determine if the theory applies only to DEER or to a wider range of algorithms.
- It would be helpful to include some comparisons to other methods; i.e., what theoretical guarantees exist about the link between certain system quantities and the corresponding optimization landscape. I see you mentioned in lines 272–273 that there are no prior works on LLE, but I assume there may be some works that link it to other system quantities. Without such comparisons, it’s hard to evaluate the strength of the results.
- The paper feels organized weirdly. Putting the related work section in the middle throws off the flow and makes it harder to follow.

---

> ### Author Rebuttal · Authors · 2025-07-31
>
> We thank the reviewer for their constructive comments. We address each point, and have incorporated all changes in the revised manuscript. We believe we have significantly strengthened our submission in doing so.
>
> ## Generality beyond DEER
>
> > will help determine if the theory applies only to DEER or to a wider range of algorithms
>
> We thank the reviewer for emphasizing the importance of highlighting the generality of our results.
>
> Theorem 2 establishes a relationship between the conditioning of the merit function and the Largest Lyapunov Exponent (LLE) $\lambda$. Specifically, we show that the Polyak–Łojasiewicz (PL) constant $\mu$ of the merit function increases monotonically with $\lambda$. The PL constant $\mu$ quantifies the conditioning of the optimization problem—i.e., how "well-conditioned" the merit function landscape is—while the LLE $\lambda$ measures the predictability of the underlying dynamical system.
>
> Theorem 2 thus demonstrates that systems with low predictability ($\lambda > 0$) exhibit poorly conditioned merit function landscapes, whereas more predictable systems ($\lambda < 0$) yield well-conditioned landscapes.
>
> Importantly, Theorem 2 concerns *the landscape of the merit function itself*, independent of any specific optimization algorithm. Therefore, our theory is not limited to Newton’s method (DEER), but extends to a broad class of optimization methods.
>
> For instance, as discussed on line 131, the PL condition guarantees that gradient descent achieves a linear convergence rate. However, the rate constant depends on the value of $\mu$, which we becomes prohibitively small in the case of unpredictable systems—leading to slow convergence. This highlights that our theoretical results are not limited to Gauss-Newton method (DEER) but also extend to gradient descent. We have clarified this point in the revised manuscript to better emphasize the broad applicability of our contributions across optimization algorithms.
>
> ## The degree of predictability has a monotonic relationship with the conditioning of the merit function landscape
>
> > How does the degree of predictability (i.e., sensitivity to perturbations) affect the results?
>
> Theorem 2 (equation 9) precisely indicates how the degree of predictability $\lambda$ influences the difficulty of the optimization problem, as measured by $\mu$. The relationship is monotonic: the higher the degree of unpredictability, the more difficult the corresponding optimization problem.
>
> In the paper we particularly emphasized the sharp change that occurs at the transition between predictability and unpredictability (at $\lambda=0$). We did so because the $\exp(\lambda T)$ in the denominator of equation 9 grows exponentially for positive $\lambda$. We have revised our paper to more heavily emphasize that the degree of predictability affects the difficulty of the optimization problem, and that the particular functional form of this relationship is what makes the predictability/unpredictability transition so stark.
>
> The effect of the degree of predictability also becomes more clear in our experimental evaluations with other methods that we have added, thanks to the reviewer’s feedback.
>
> ## Strengthened Experimental Evaluation
>
> We have strengthened our experimental evaluations by including results for both **gradient descent** and a **quasi-Newton method**. Specifically, we revisited our experiment involving the mean-field RNN (see Figure 2 in the paper). In the setting depicted in the rightmost plot of this figure (state dimension $D = 100$, sequence length $T = 954$), we also measured the rates of convergence of a quasi-Newton method and gradient descent. The results are summarized in the following table:
>
> ###  Iterations until convergence
> | LLE               |   Newton |   quasi-Newton |   gradient descent |
> |:------------------|---------:|---------------:|-------------------:|
> | -0.7              |        3 |             10 |                 31 |
> | -0.5              |        4 |             13 |                 58 |
> | -0.35             |        4 |             17 |                206 |
> | -0.2              |        4 |             26 |                435 |
> | -0.1              |        4 |             66 |                954 |
> | -0.1 to 0         |       18 |            674 |                954 |
> | 0 to 0.05         |      147 |            954 |                954 |
> | 0.05 to 0.1       |      642 |            954 |                954 |
> | 0.1 to 0.14       |      895 |            954 |                954 |
> | greater than 0.14 |      954 |            954 |                954 |
>
>
> This table clarifies two points raised in the review:
>
> 1. The effect of the degree of predictability (the LLE) on rate of convergence.
>
> For all 3 optimizers, the number of steps to convergence increases monotonically with the value of the LLE. This is the case because, as we proved in Theorem 2, the conditioning $\mu$ of the merit function landscape degrades monotonically as the LLE $\lambda$ increases.
>
> 2. The applicability of our theory across optimizers.
>
> We observe that this monotonic relationship between LLE and numbers of steps to converge holds across optimizers, thus strengthening our evaluation as per the reviewer’s recommendation. Newton’s method converges in a small number of steps all the way up to the threshold between predictability and unpredictability (LLE= 0). Intuitively, the performance of the other optimizers degrades more quickly as unpredictability increases because quasi-Newton and gradient descent use less information about the curvature of the loss landscape.
>
> ## Comparisons with Related Work
>
> >  the related works section
>
> We agree about the location of related works. We have moved it to the conclusion to improve the flow.
>
> > comparison to other methods
>
> To the best of our knowledge, our work is novel in connecting the LLE of a dynamical system with the conditioning of its merit function landscape. Nonetheless, we thank the reviewer for highlighting the need to expand the comparison of our contributions with the literature. We have done so in our revision, better contextualizing the strength of our results.
>
> Other system-theoretic quantities have been proposed to assess the “parallelizability” of dynamical systems. Particularly relevant to our work is the study we cited by Chartier and Philippe (reference [21] in the paper), which investigates the parallelizability of input-free continuous-time systems using a multiple-shooting approach. Their analysis centers on the *logarithmic norm* of the system Jacobian, a quantity that can be roughly interpreted as a “one-step” analogue of the LLE. Specifically, while the LLE captures the *asymptotic* growth rate of perturbations over time, the logarithmic norm quantifies the *instantaneous* contractivity of the system’s Jacobian. As such, a negative logarithmic norm implies a negative LLE, but the converse does not necessarily hold. In this way, our work is a significant generalization of Chartier and Philippe.
>
> We also surveyed literature in theoretical computer science relating to the parallelization of nonlinear recursions. In particular, we cited Hyafil and Kung (ref [17]) and Kung (ref [18]). They showed that when the nonlinear recursion is a rational function (i.e., the ratio of two polynomials), and if the degree of that rational function is greater than one, then such a nonlinear recursion does not benefit from parallelization as the number of processors grows (cf. Theorem 4 of [17]). However, our work greatly generalizes this foundational contribution, because our formulation in terms of the LLE applies to *any* recursion, not just rational functions.
>
> ## Additional clarifications
>
> > How does the system’s dimension impact the results?
>
> Theorem 2 does not explicitly depend on the system dimension $D$, just the LLE $\lambda$. However, the LLE itself can depend on the system dimension. For example, consider a linear dynamical system with transition matrix $A$. If the entries of $A$ are drawn i.i.d from a normal distribution with mean 0 and variance $g$, then classical results from random matrix theory (e.g., Sompolinsky, et al. "Chaos in random neural networks”) show that, as the dimension $D$ grows, the largest eigenvalue scales like $g\sqrt{D}$. Consequently, if $g$ is not appropriately scaled with dimension, the LLE may increase and eventually become positive as $D$ increases.
> > the example in lines 177–182
>
> In this example, we provide intuition as to why unpredictable systems have flat merit landscapes. We used two different ideas: the *residuals* and the *errors*.
>
> We define the residual $\mathbf{r}(\mathbf{s})$ in equation 2. The residual at time $t$ is $s_t - f(s_{t-1})$. The residual is a function only of our current guess for the sequence trajectory $\mathbf{s}$.
>
> We have clarified the definition of “errors” in our revision. The error is the distance between our current guess for the sequence $\mathbf{s}$ and the true sequence trajectory $\mathbf{s*}$. More precisely, we can define the error at time $t$ as $e_t(s_t, s_t*) = | s_t - s_t* |$. In contrast to the residual, the error is a function both of our current guess $\mathbf{s}$ and of the true trajectory $\mathbf{s*}$.
>
> This example shows that for an unpredictable system, small residuals do *not* necessarily imply small errors.
>
> In this example, we do assume that the residual for all times greater than 1 is 0, that is $s_t = f(s_{t-1})$ for $t > 1$. However, we do not state that the “residuals grow over time.” Instead, we note that the *error* (that is, the difference between $s_t*$ and $s_t$) grows over time (see lines 180-182). This exponential growth in the error—in spite of the small and bounded value of the residual—highlights why many trajectories can have small residual for unpredictable systems, even though they may be very different from the true trajectory $\mathbf{s*}$. This is why unpredictable systems have flat merit landscapes.

---

### Note · Authors · 2025-08-15

We thank the reviewers for their time and effort!

We were heartened by the positive response to our contributions. Reviewers praised the creativity, practicality, and novelty of our work, as well as the clarity of the presentation. All four reviewers recommended acceptance and said our rebuttal fully addressed their questions.

We now recap our main contributions and summarize improvements.

## Main contribution

Our paper answers the question: **when can a nonlinear state space model be efficiently parallelized?** Sequential processes pervade machine learning, yet modern accelerators favor parallel computation.

Recent advances (i.e., DEER, Lim et al.) reformulate sequential evaluation as a parallel optimization that is fast when it converges in a few steps. However, DEER can be slower than sequential evaluation when the optimization takes many steps to converge. The factors that govern the difficulty of these optimization problems were unclear, limiting the larger adoption of the technique.

Our central result is that predictable (e.g. contracting) systems yield well-conditioned optimization problems that can be parallelized efficiently. Unpredictable (e.g. chaotic) systems produce poorly conditioned optimization problems and are not easily parallelizable. We formalize and prove this (Theorem 2) and validate it empirically.

## Improvements from reviewer feedback

The main feedback from reviewers was that we should emphasize our contribution’s generality and practicality.

To show generality, we added experiments beyond Gauss–Newton (DEER), demonstrating that our theory also holds for gradient descent and a quasi-Newton method.

To show practicality, we reported wall-clock results. In predictable settings, parallel evaluation achieves order-of-magnitude speedups; in unpredictable settings, it is order-of-magnitude slower than sequential evaluation. We clarified computation of the Largest Lyapunov Exponent (LLE) that quantifies predictability and expanded discussion of how our paper can guide design of nonlinear sequence models.

These additions underscore the generality and practicality of our contributions. We believe our paper will be of great interest to the NeurIPS community. Thank you reviewers and AC for your hard work in the review process!

---

### Decision · Program_Chairs · 2025-09-17

**Decision:**

Accept (poster)

**Comment:**

This paper presents a theoretical framework demonstrating that a system’s predictability governs the number of optimization steps required for evaluation. This relationship is particularly significant for assessing the efficiency of parallel computing. A key strength of the work lies in its central contribution of linking the Largest Lyapunov Exponent to PL conditioning in optimization problems. The formalization of predictable and unpredictable systems is also creative and provides a strong foundation for the authors’ main results. As most of the reviewers’ comments have been adequately addressed, this paper is suitable for publication.